# mTORC1 is necessary but mTORC2 and GSK3β are inhibitory for AKT3-induced axon regeneration in the central nervous system

**Linqing Miao[1†], Liu Yang[1†], Haoliang Huang[1], Feisi Liang[1], Chen Ling[2], Yang Hu[1,3*]**

[1]Shriners Hospitals Pediatric Research Center, Temple University Lewis Katz School of Medicine, Philadelphia, United States; [2]Division of Cellular and Molecular Therapy, Department of Pediatrics, University of Florida College of Medicine, Gainesville, United States; [3]Department of Anatomy and Cell Biology, Temple University Lewis Katz School of Medicine, Philadelphia, United States

**Abstract** Injured mature CNS axons do not regenerate in mammals. Deletion of PTEN, the negative regulator of PI3K, induces CNS axon regeneration through the activation of PI3K-mTOR signaling. We have conducted an extensive molecular dissection of the cross-regulating mechanisms in axon regeneration that involve the downstream effectors of PI3K, AKT and the two mTOR complexes (mTORC1 and mTORC2). We found that the predominant AKT isoform in CNS, AKT3, induces much more robust axon regeneration than AKT1 and that activation of mTORC1 and inhibition of GSK3β are two critical parallel pathways for AKT-induced axon regeneration. Surprisingly, phosphorylation of T308 and S473 of AKT play opposite roles in GSK3β phosphorylation and inhibition, by which mTORC2 and pAKT-S473 negatively regulate axon regeneration. Thus, our study revealed a complex neuron-intrinsic balancing mechanism involving AKT as the nodal point of PI3K, mTORC1/2 and GSK3β that coordinates both positive and negative cues to regulate adult CNS axon regeneration.

*For correspondence: yanghu@temple.edu

[†]These authors contributed equally to this work

**Competing interests:** The authors declare that no competing interests exist.

## Introduction

Injuries of mature central nervous system (CNS) axons result in loss of vital functions due to the failure of CNS axons to regenerate (*Schwab and Bartholdi, 1996*; *Goldberg et al., 2002b*; *Fitch and Silver, 2008*). We and other investigators have found that the activation of phosphatidylinositol 3-kinase (PI3K) by the deletion of phosphatase and tensin homolog (PTEN) induces CNS axon regeneration through the activation of mammalian target of rapamycin (mTOR) signaling (*Park et al., 2008*; *Liu et al., 2010*). PI3K is a lipid kinase, which can be activated by receptor tyrosine kinase (RTK) and subsequently phosphorylates phosphatidylinositol 4,5-bisphosphate ($PIP_2$) in the lipid membrane to produce phosphatidylinositol (3,4,5)-triphosphate ($PIP_3$). $PIP_3$ in turn recruits AKT, a member of the AGC family of serine/threonine kinases, to the membrane to be activated by phosphorylation at T308 via phosphoinositide-dependent kinase-1 (PDK1) (*Manning and Cantley, 2007*). PTEN is a lipid phosphatase that converts $PIP_3$ to $PIP_2$, and thus inhibits the activation of downstream effectors of PI3K. The PI3K-AKT pathway is the main effector by which RTKs promote cell survival and growth in response to growth factor signaling (*Song et al., 2012a*). One of the multiple AKT downstream effectors is the tuberous sclerosis complex (TSC1/TSC2), the negative regulator of mTOR complex 1 (mTORC1). Thus AKT activation removes the inhibition of TSC and activates mTORC1. The PI3K-

**eLife digest** The central nervous system consists of the neurons that make up the brain and spinal cord. An important part of a neuron is the long, slender projection along which electrical signals travel, called the axon. In the central nervous system of mammals, damaged axons cannot regrow, which is why spinal injuries or optic nerve injuries can result in life-long neuronal deficits.

Recent studies have found that activating a particular signaling pathway in central nervous system neurons causes their axons to regenerate. A key protein in this pathway is called AKT. Several signaling cascades are triggered by AKT to regulate cell survival and growth, but it was not known how the different branches of the AKT pathway are involved in axon regeneration.

Miao, Yang et al. have now investigated AKT's role in axon regeneration using a range of approaches to manipulate signaling in damaged mouse neurons. This revealed that a particular form of AKT (called AKT3) causes damaged axons to regenerate to a greater extent than other forms of this protein. This response depends on two parallel pathways: one in which AKT3 activates a protein complex called mTORC1, and one where AKT3 inhibits a protein called GSK3β. In addition, another protein complex called mTORC2, which is closely related to mTORC1, helps to inhibit the activity of AKT3 on GSK3β and hence inhibits axon regeneration.

These findings reveal that a complex balancing mechanism, with AKT at its center, coordinates the many signals that regulate axon regeneration. Future studies into this system could ultimately help to develop new treatments for brain and spinal injuries.

AKT-mTORC1 pathway is a master regulator of protein synthesis and cellular growth (*Manning and Cantley, 2007*; *Laplante and Sabatini, 2012*).

The two best characterized substrates of mTORC1 have been suggested as mediators of mTORC1's prominent roles in regulating cell growth, size, proliferation, motility and survival. One of these is the eukaryotic initiation factor 4E-binding proteins (4E-BPs); their phosphorylation releases inhibition of eukaryotic translation initiation factor 4E and initiates cap-dependent translation. The other is the ribosomal protein S6 kinases (S6Ks); their phosphorylation is critical for mRNA biogenesis, translation initiation and elongation (*Hay and Sonenberg, 2004*). Phosphorylation of ribosome protein S6 (pS6) by S6K has often served as a marker for mTORC1 activation. Interestingly, S6K also functions as feedback inhibition of RTK/PI3K signaling, which balances mTORC1 activation (*Radimerski et al., 2002*; *Um et al., 2004*; *Yang et al., 2014*). Both 4E-BP inhibition and S6K activation promote protein synthesis, but 4E-BPs act specifically to control cell proliferation (cell number) and S6Ks preferentially regulate cell growth (cell size) (*Dowling et al., 2010*; *Ohanna et al., 2005*). We previously reported that, although S6K1 activation, but not 4E-BP inhibition, is sufficient for axon regeneration, 4E-BP inhibition is necessary for PTEN deletion-induced axon regeneration (*Yang et al., 2014*). Phosphorylation and inhibition of another AKT substrate, glycogen synthase kinase 3β (GSK-3β), is critical for neuronal polarization, axon branching and axon growth (*Kim et al., 2011b*). However, how GSK-3β regulates peripheral axon regeneration is controversial (*Saijilafu et al., 2013*; *Zhang et al., 2014a*; *Gobrecht et al., 2014*) and its function in CNS axon regeneration remains to be determined.

Although the mechanism is unclear, PI3K also activates mTOR complex 2 (mTORC2) in a ribosome-dependent manner (*Zinzalla et al., 2011*), which in turn phosphorylates AKT at S473 (pS473) (*Sarbassov et al., 2005*; *Guertin et al., 2006*; *Hresko and Mueckler, 2005*). pS473 enhances phosphorylation of AKT-T308 (*Yang et al., 2002*; *Scheid et al., 2002*), and inhibition of S473 phosphorylation by destroying mTORC2 decreases AKT-T308 phosphorylation (*Sarbassov et al., 2005*; *Carson et al., 2013*; *Yuan et al., 2012*; *Hresko and Mueckler, 2005*; *Guertin et al., 2009*). mTORC2 is involved in cell survival and actin cytoskeleton dynamics (*Jacinto et al., 2004*) and, like mTORC1, plays a role in lipogenesis and adipogenesis (*Yao et al., 2013*; *Lamming and Sabatini, 2013*). It is not clear how these two mTOR complexes interact to determine multiple cellular events, and this information is particularly lacking for axon regeneration.

AKT appears to be the nodal point that acts as the key substrate of both PI3K-PDK1 and PI3K-mTORC2, and is also the critical upstream regulator of mTORC1 and GSK3β. By exploiting the

anatomical and technical advantages of retinal ganglion cells (RGCs) and the crushed optic nerve (ON) as an in vivo model, we elucidated the important roles of AKT, mTORC1/2 and GSK3β in adult CNS axon regeneration. Understanding this cross-regulating mechanism should provide promising therapeutic targets for CNS injuries.

## Results

### Three AKT isoforms display different effects on axon regeneration and RGC survival

To determine the role of AKT in axon regeneration, we generated adeno-associated virus 2 (AAV2) vectors containing Myr-3HA-AKT1, 2 and 3 to express membrane-bound constitutively active forms of HA-tagged AKTs in vivoWe and other investigators have demonstrated a specific tropism of AAV2 for RGCs after intravitreal injection (*Park et al., 2008*; *Pang et al., 2008*; *Hu et al., 2012*; *Boye et al., 2013*; *Yang et al., 2014*). We made AAV2-AKT viruses containing triple mutant capsid (Y444, 500, 730F) to take the advantage of the high RGC transduction efficiency of capsid-mutated AAV2 (*Petrs-Silva et al., 2011*): transduction exceeded 80% of RGC, based on the ratio of HA (transgene tag) to Tuj1 (antibody for RGC marker, β-III tubulin) positive cells in flat-mount retinas (*Figure 1A,B*). We also confirmed that all three isoforms of AKT produced the expected changes in RGCs: significantly increased levels of pAKT-S473 and pS6 (mTORC1 activation marker), indicating activation of AKT and mTORC1 (*Figure 1A,B*, note that pAKT-T308 antibody did not effectively immunostain retina). Western blot analysis of retina lysates showed comparable expression of AKT isoforms (HA levels), but levels of pT308 and pS6 were significantly higher in retinas transfected by AKT3 than by AKT1 (*Figure 1C,D*). This difference indicates the greater activity of AKT3 in retina. We noticed that both pAKT-T308 and pAKT-S473 antibodies can only be used to detect AKT1 and AKT3 reliably in Western blot, whereas the pAKT2-S474 specific antibody readily detected p-AKT2 (*Figure 1C*).

We then performed ON crush in wild type (WT) mice 2 weeks after intravitreal injection of these AAV-AKTs. RGC axons that regenerated through the lesion site were labeled anterogradely by intra-vitreal injection of the tracer Alexa 488-conjugated cholera toxin β (CTB), and imaged and quantified in ON longitudinal sections at 2 weeks post-crush (*Figure 2—figure supplement 1*). Interestingly, the three isoforms of AKT elicited different patterns of axon regeneration: AKT2 and AKT3 caused significantly more axon regeneration than AKT1; AKT2 induced the longest axon growth (*Figure 2A, B*). The differing capabilities of the three isoforms of AKT in axon regeneration could not be explained simply by their expression levels, which HA staining showed to be comparable (*Figure 1C,D*). The higher pT308 and pS6 levels and more potent axon regeneration induced by AKT3 suggest that its function in retina differs from those of AKT1 and AKT2. Similar to their effects on axon regeneration, all three isoforms increased RGC survival to more than 40% based on Tuj1 staining in retinal flat-mounts, at least a 1-fold increase over WT mice; and significantly more RGCs survived in mice injected with AAV-AKT3 than with AKT1 (*Figure 2C,D*). This series of experiments demonstrated that the downstream effector of PI3K, AKT, promotes both RGC survival and ON regeneration, and that there are significant differences among the three AKT isoforms.

### AKT1 and AKT3 are the predominant isoforms of AKT in RGCs

Consistent with a previous report (*Yang et al., 2015*), in situ hybridization and Western blot detected all three isoforms of AKT in retina (*Figure 2—figure supplement 2A,B*). Since AKT3 is the predominant isoform in adult mouse brain (50% AKT3, 30% AKT1 and 20% AKT2) (*Easton et al., 2005*), we determined whether this is also the case in RGCs. We employed the RiboTag mice, which are generated by knocking in the HA-tagged ribosome protein Rpl22 (Rpl22[HA]) to the endogenous Rpl22 allele, immediately after the floxed endogenous Rpl22 gene (*Sanz et al., 2009*). After intravitreal injection of AAV2-Cre, Rpl22[HA] was expressed specifically in RGCs after Cre-mediated deletion of endogenous Rpl22 (*Figure 2—figure supplement 2C*). This allowed us to selectively immunoprecipitate (IP) RGC-specific ribosomes (Ribo-IP) in situ (*Doyle et al., 2008*; *Heiman et al., 2008*; *Sanz et al., 2009*) and acquire high quality RGC-specific ribosome-associated translating mRNA from mouse retinas (*Figure 2—figure supplement 2D*). We then used three biological replicates of RGC-translating mRNAs from WT mice (8–10 mice/replicate) to perform RNA deep sequencing;

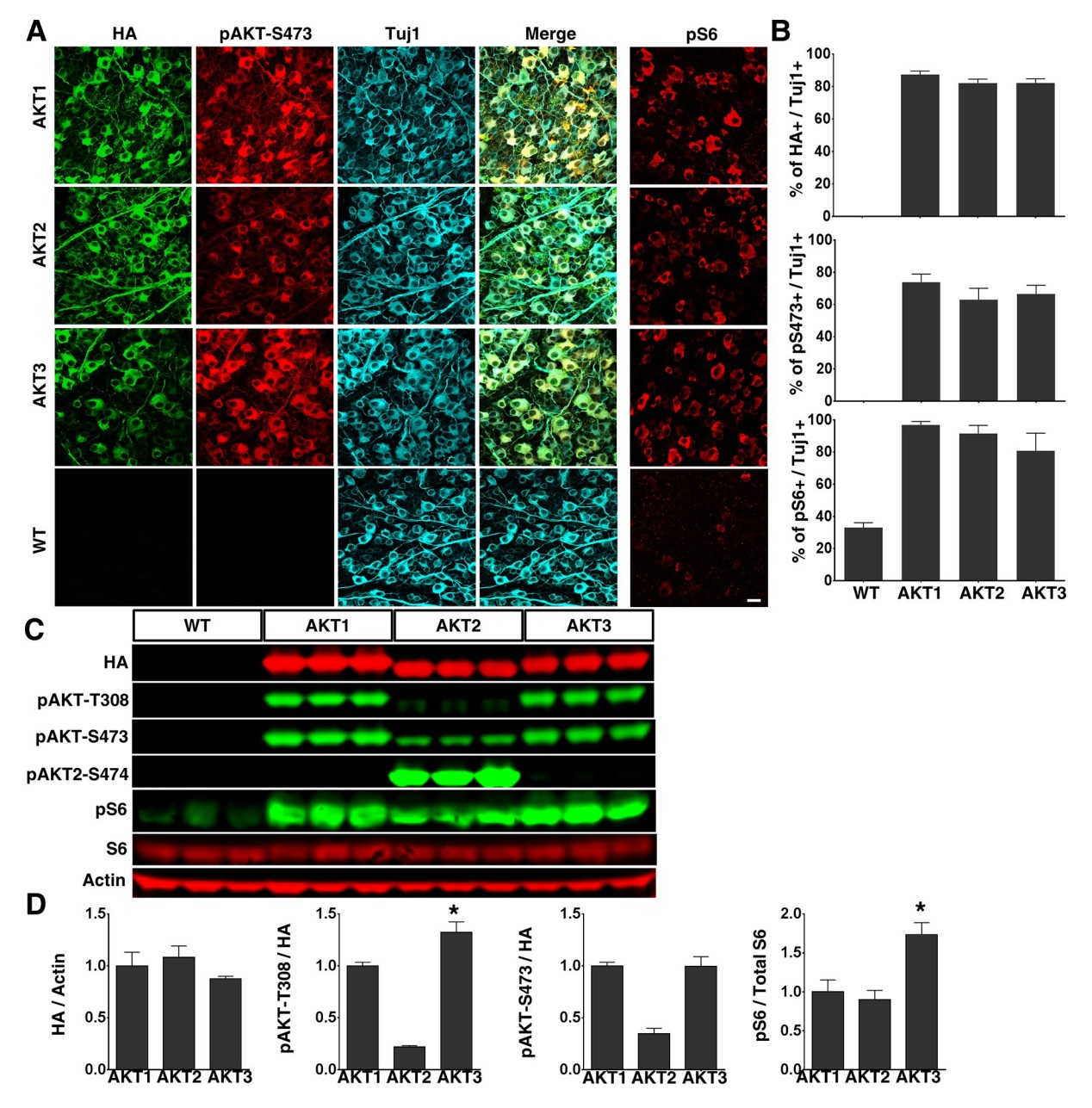

**Figure 1.** Overexpression of three constitutively active AKT isoforms in RGCs. (**A**) Confocal images of flat-mounted retinas showing co-labeling of HA tag, Tuj1, pAKT-S473 and their merged images, and pS6 in a separate retina sample. Scale bar, 20 μm. (**B**) Quantification of HA, pAKT-S473 or pS6 positive RGCs. Data are presented as means ± s.e.m, n=6. (**C**) Western blot of retina lysates from three biological replicates showing expression levels of HA-AKT isoforms, and phosphorylation levels of AKT-T308, AKT-S473 and S6. (**D**) Quantification of Western blots. *p<0.05. Data are presented as means ± s.e.m, n=3.

each library acquired about 25 million reads and 93% of total reads aligned to unique genes in the mouse genome (mm9) by RUM (*Grant et al., 2011*). Measurement of transcript abundance in fragments per Kb of exon per million fragments mapped (FPKM) (*Mortazavi et al., 2008*) indicated that AKT1 and AKT3 were the two major isoforms of AKT in RGCs; each represented 45% of total AKT, a vivid contrast to AKT2 (10%) (*Figure 2—figure supplement 2E*).

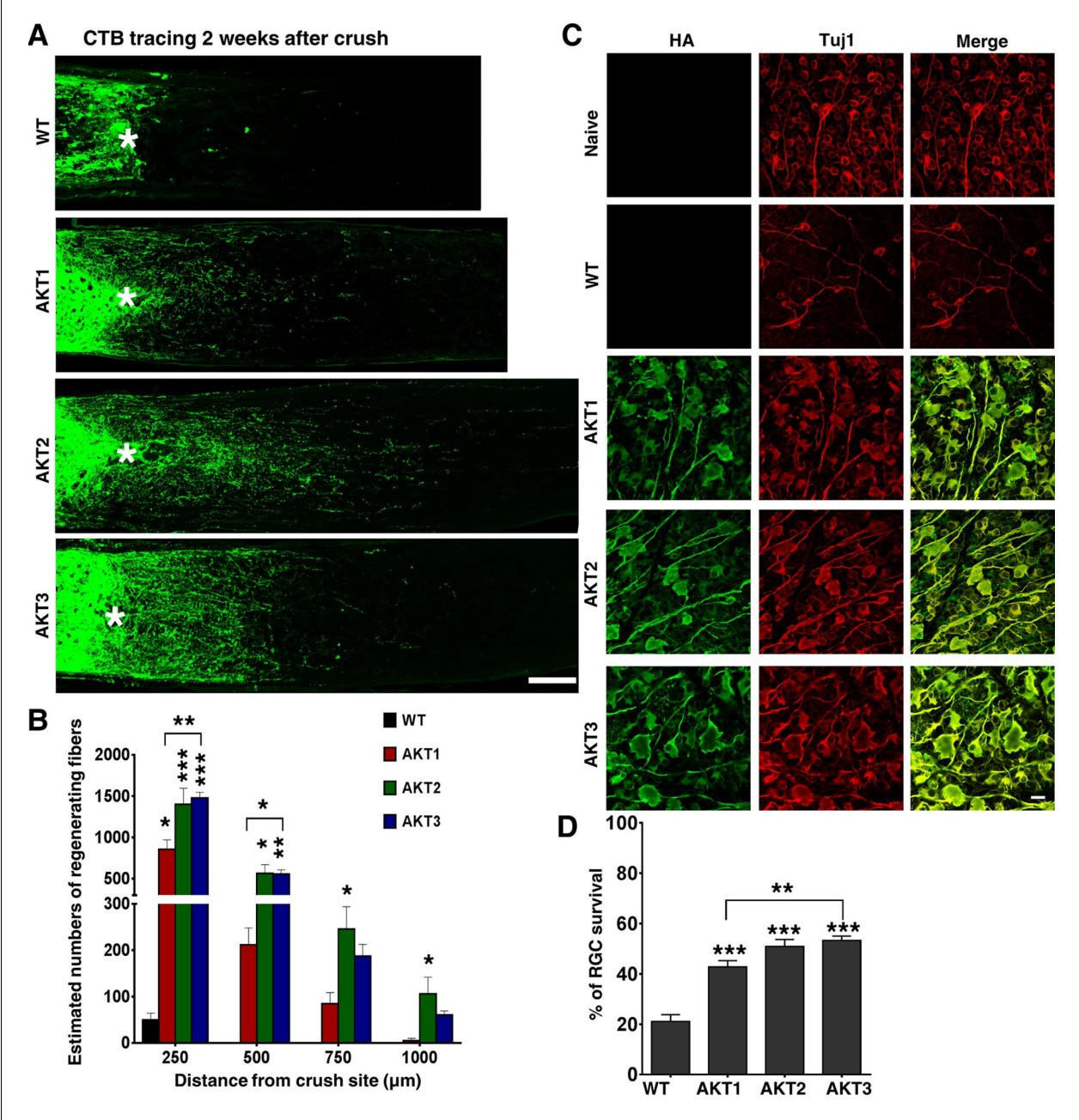

**Figure 2.** Differential effects of three AKT isoforms on axon regeneration and RGC survival. (**A**) Confocal images of ON longitudinal sections showing regenerating fibers labeled with CTB-Alexa 488 2 weeks after ON crush. Scale bar, 100 μm. *crush site. (**B**) Quantification of regenerating fibers at different distances distal to the lesion site. Data are presented as means ± s.e.m, n=10–20. (**C**) Confocal images of flat-mounted retinas showing co-labeling of HA-AKTs and Tuj1, 2 weeks after ON crush. Scale bar, 20 μm. (**D**) Quantification of surviving RGCs, represented as percentage of Tuj1 positive RGCs in the injured eye, compared to the intact contralateral eye. Data are presented as means ± s.e.m, n=8–10. *p<0.05, **p<0.01, ***p<0.001.

The following figure supplements are available for figure 2:

**Figure supplement 1.** Illustration of the regenerating axon quantification procedure.

**Figure supplement 2.** Endogenous expression of three AKT isoforms in RGCs.

# The opposite roles of phosphorylation of AKT-T308 by PDK1 and phosphorylation of AKT-S473 by mTORC2 on axon regeneration and GSK3β phosphorylation

AKT1 and AKT3 are expressed equally in RGCs, but AKT3 is activated more potently in retina and induces better axon regeneration than AKT1. We therefore focused on AKT3 to investigate which of its domains is critical for axon regeneration. We generated three AKT3 mutants and verified their activities in RGCs after AAV infection: AKT3 kinase dead (KD) mutant (K177M), T305A/S472A (AA) mutant (equal to T308 and S473 in AKT1, T308 and S473 are used throughout the paper for convenience unless AKT3 is specified) and single T305A mutant were all equally expressed in RGCs and, as we expected, all three mutants showed significantly lower phosphorylation of T308, S473 and S6 than WT AKT3 (*Figure 3*). Consistent with their loss of AKT kinase function, the three mutants did not induce axon regeneration (*Figure 4A,B*), indicating that the phosphorylation of AKT-T308 and the kinase activity of AKT are critical for axon regeneration.

We were surprised to find, however, that the AKT3-S472A mutant produced a small but statistically significant increase in axon regeneration at 500 μm distal to the crush site and a trend toward increase at other distances, compared to WT AKT3 (*Figure 4C,D*), suggesting that pS473 inhibits the effect of AKT on axon regeneration. This contradicts the common theme that phosphorylation of both T308 and S473 is additive to AKT activity (*Bhaskar and Hay, 2007*; *Yang et al., 2002*; *Scheid et al., 2002*). To confirm this unanticipated finding, we tested whether mTORC2 also is inhibitory for AKT-induced axon regeneration, since S473 of AKT is phosphorylated by mTORC2 (*Sarbassov et al., 2005*; *Guertin et al., 2006*; *Hresko and Mueckler, 2005*). We blocked mTORC2 genetically by deleting its essential component, *Rictor* (rapamycin-insensitive companion of mTOR) (*Guertin et al., 2006*; *Laplante and Sabatini, 2012*). We injected AAV-AKT3 and AAV-Cre together into one eye of *Rictor* floxed mice and compared ON regeneration to that elicited by the injection of AAV-AKT3 alone into the contralateral eye. Consistent with our AKT3-S472A result, AKT3 and *Rictor* KO produced even more extensive and lengthier axon regeneration than AKT3 alone (*Figure 4C,D*). This was not due to an additive effect because *Rictor* KO alone did not cause any axon regeneration (*Figure 4—figure supplement 1*), but presumably through inhibition of S473 phosphorylation. Indeed, both immunostaining and Western blot analysis confirmed that AKT3-S472A mutant and *Rictor* deletion significantly decreased pS473 levels (*Figure 5A,B,F*). The enhanced axon regeneration was not due to increased expression of AKT3, which was slightly decreased in *Rictor* KO mice (*Figure 5B,D*). It was also not due to enhanced activation of AKT or mTORC1, because levels of pT308 and pS6 were also decreased (*Figure 5B,E,G*). This is consistent with the positive effect of pS473 on T308 phosphorylation (*Sarbassov et al., 2005*; *Carson et al., 2013*; *Yuan et al., 2012*; *Hresko and Mueckler, 2005*; *Guertin et al., 2009*; *Yang et al., 2002*; *Scheid et al., 2002*). Surprisingly, mTORC2 and pS473 had a negative effect on GSK3β-S9 phosphorylation; deletion of *Rictor* significantly increased pGSK3β-S9 level (*Figure 5B,H*). AKT3-S472A mutant also increased GSK3β-S9 phosphorylation although with large variation. *Rictor* KO alone did not increase pGSK3β-S9 compared to WT mice (*Figure 5C*), suggesting that AKT kinase activity is required for GSK3β phosphorylation. Since GSK3β-S9 phosphorylation has been suggested to promote peripheral axon regeneration (*Saijilafu et al., 2013*; *Zhang et al., 2014a*), inhibition of AKT-S473 phosphorylation may increase CNS axon regeneration through enhanced inactivation of GSK3β. Taken together, our results demonstrate that, in contrast to pAKT-T308, mTORC2 and pAKT-S473 inhibit GSK3β phosphorylation, which may contribute to their negative effect on axon regeneration.

Since retina flat-mount preparations revealed HA signals in RGC cell bodies as well as proximal axons, we investigated the distribution of AKT in ON. All three AKT isoforms and AKT3-S472A mutant were translocated into ON. Intriguingly however, the distribution in ON of the three AKT3 mutants (KD, AA and T305A) that did not induce axon regeneration was significantly limited (*Figure 4—figure supplement 2A*), suggesting a local function of axonal AKT in axon regeneration. Consistent with this idea, AKT3 was also present in regenerating axons (*Figure 4—figure supplement 2B*).

Consistent with their loss of kinase function, the three AKT3 mutants (AKT3-KD, AA and T305A mutants) significantly decreased survival of axotomized RGCs compared to WT AKT3 (*Figure 4—figure supplement 3*). Neither the AKT3-S472A mutant nor *Rictor* deletion significantly changed RGC

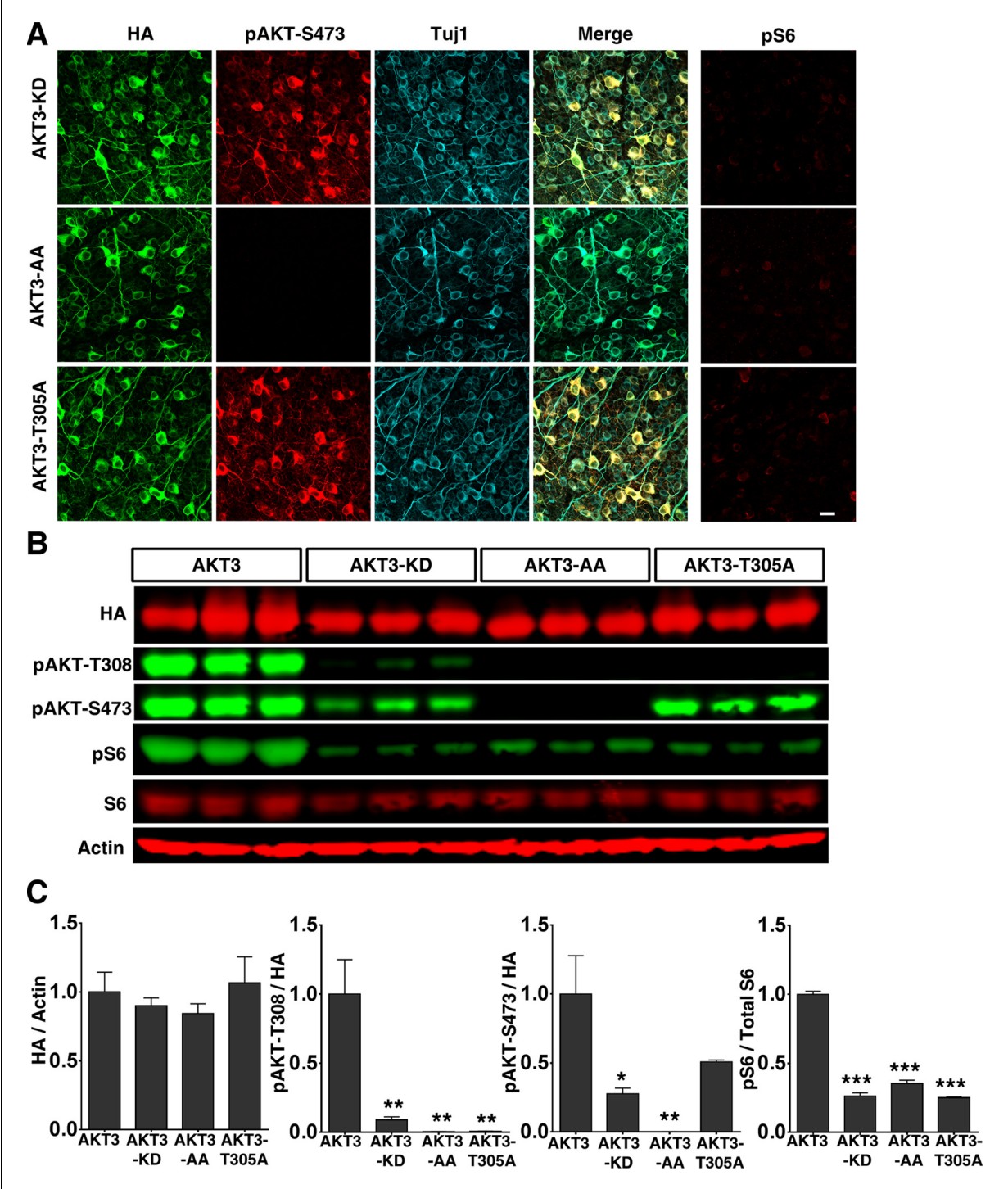

**Figure 3.** Over expression of three AKT3 mutants in RGCs. (A) Confocal images of flat-mounted retinas showing co-labeling of HA tag, Tuj1, pAKT-S473 and their merged images, and pS6 in a separate retina sample. Scale bar, 20 μm. (B) Western blot of retina lysates from three biological replicates showing expression level of HA-AKT, and phosphorylation level of AKT-T308, AKT-S473 and S6. (C) Quantification of Western blots. *p<0.05, **p<0.01, ***p<0.001. Data are presented as means ± s.e.m, n=3.

survival (*Figure 4—figure supplement 3*), indicating that increased axon regeneration is not due to increased neuron survival.

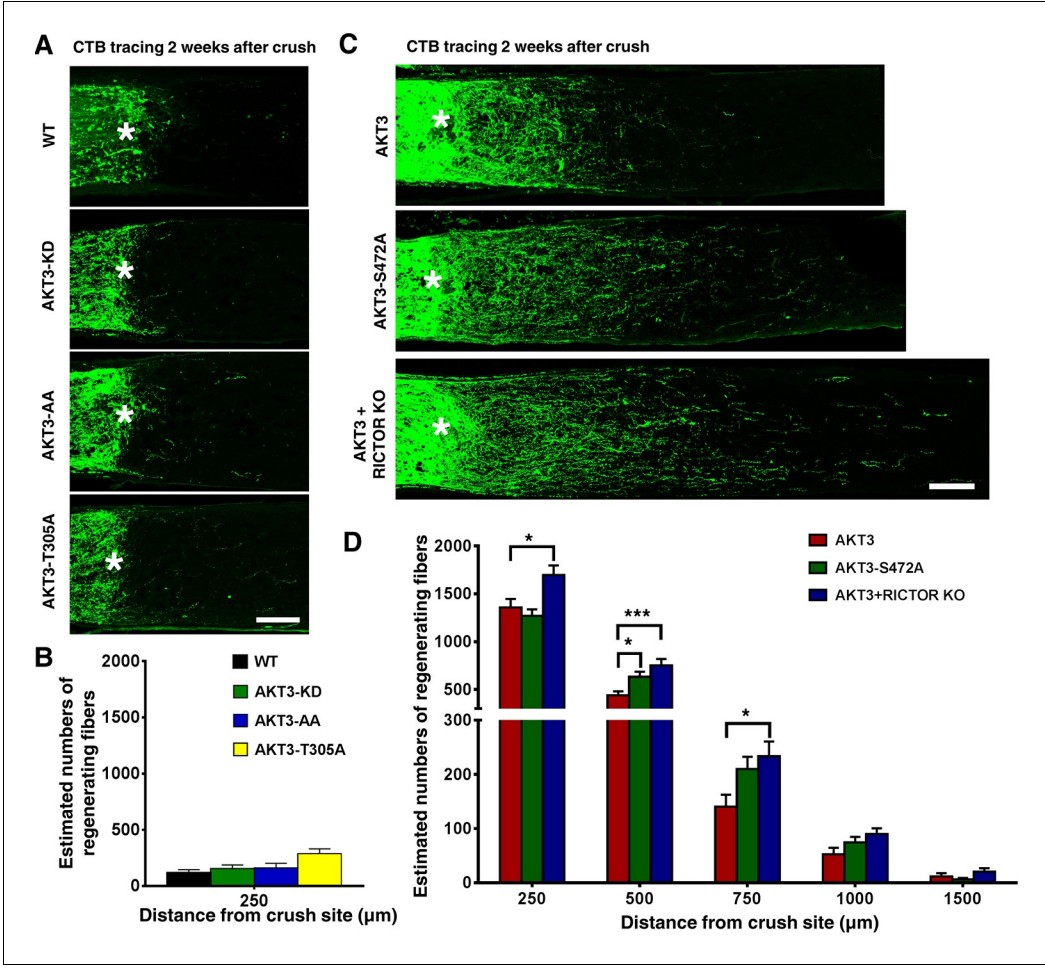

**Figure 4.** Phosphorylation of AKT3-T305 is necessary but phosphorylation of AKT3-S472 by mTORC2 is inhibitory for axon regeneration. (**A**) Confocal images of ON longitudinal sections showing lack of regenerating fibers 2 weeks after ON crush. Scale bar, 100 μm. *crush site. (**B**) Quantification of regenerating fibers at different distances distal to the lesion site. Data are presented as means ± s.e.m, n=6–8. (**C**) Confocal images of ON longitudinal sections showing regenerating fibers labeled with CTB 2 weeks after ON crush. Scale bar, 100 μm. *crush site. (**D**) Quantification of regenerating fibers at different distances distal to the lesion site. *p<0.05, ***p<0.001. Data are presented as means ± s.e.m, n=20–30.

The following figure supplements are available for figure 4:

**Figure supplement 1.** AAV-Cre-mediated RGC-specific deletion of *Rictor, Rptor, Mtor* or *Gsk3a* did not induce axon regeneration.

**Figure supplement 2.** Translocation of AKT in ON.

**Figure supplement 3.** The effects of AKT3 mutants on RGC survival.

Taken all together, these results show that AKT kinase activity is required for both axon regeneration and RGC survival; and that phosphorylation of T308 by PI3K-PDK1 and phosphorylation of S473 by mTORC2 play opposite roles in GSK3β phosphorylation/inhibition and axon regeneration.

## mTORC1 and its downstream effectors are essential for AKT3-induced axon regeneration

We next tested the influence of mTORC1 on AKT3-induced axon regeneration. Previous studies of mTORC1 function in axon regeneration relied on pharmacological inhibition of mTORC1 by

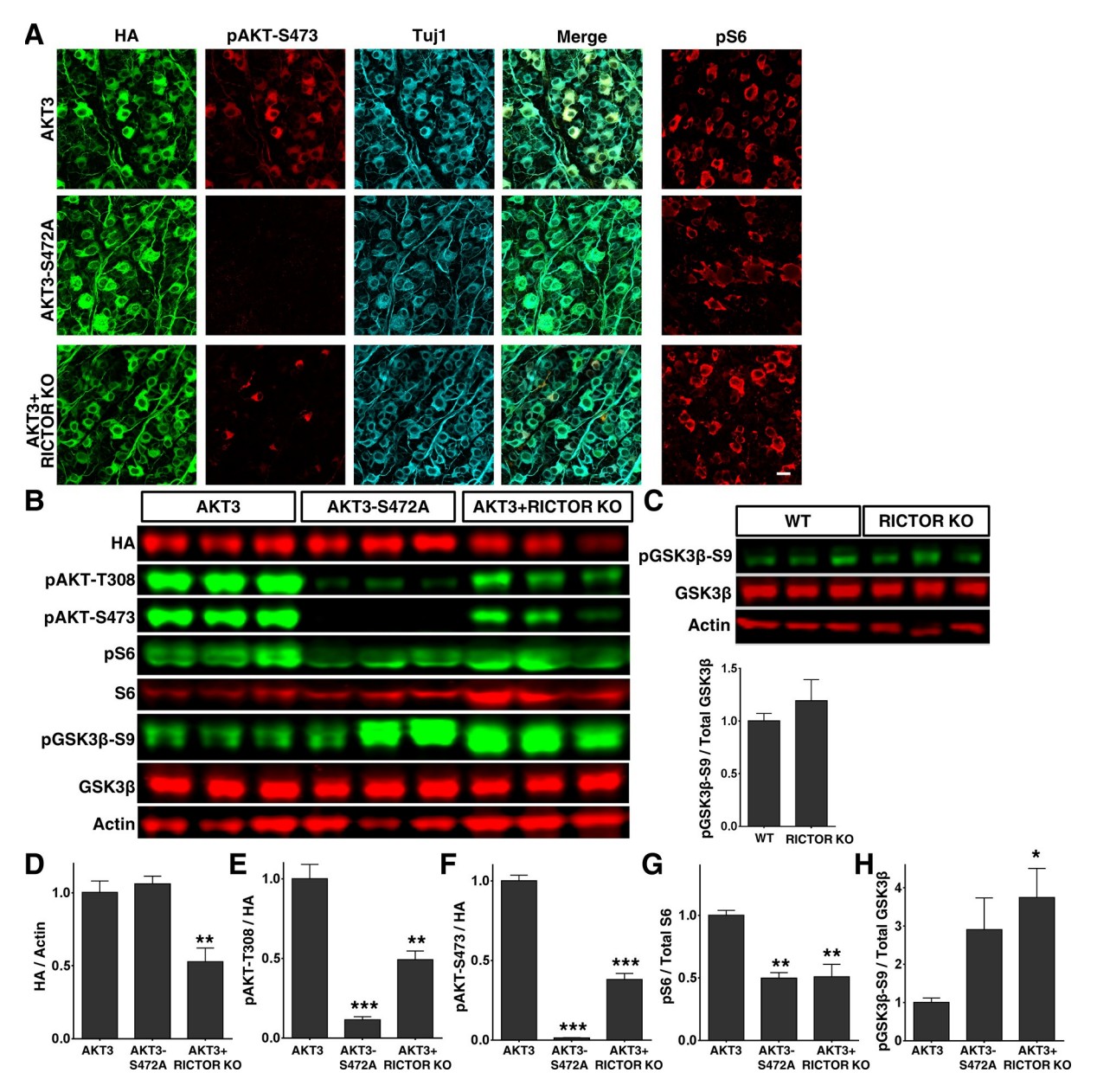

**Figure 5.** Increased phosphorylation of GSK3β-S9 after blocking AKT-S473 phosphorylation in RGCs. (**A**) Confocal images of flat-mounted retinas showing co-labeling of HA tag, Tuj1, pAKT-S473 and their merged images, and pS6 in a separate retina sample. Scale bar, 20 μm. (**B**) Western blots of retina lysates from three biological replicates showing expression level of HA, and phosphorylation levels of AKT-T308, AKT-S473, S6 and GSK3β-S9. (**C**) Western blots of retina lysates from three biological replicates showing phosphorylation levels of GSK3β-S9. (**D-H**) Quantification of Western blots. *p<0.05, **p<0.01, ***p<0.001. Data are presented as means ± s.e.m, n=3.

rapamycin, which generated inconsistent or contradictory results (*Christie et al., 2010*; *Abe et al., 2010*; *Park et al., 2008*). To definitively determine the role of mTORC1, we disrupted mTORC1 by deleting *Rptor* (regulatory associated protein of mTOR) in RGCs specifically. *Rptor* is unique to mTORC1, and its deletion totally blocks mTORC1 activity (*Laplante and Sabatini, 2012*; *Guertin et al., 2006*). We injected AAV-AKT3 and AAV-Cre together into one eye of *Rptor* floxed mice and compared the ON regeneration to that obtained with injection of AAV-AKT3 alone into the contralateral eye. *Rptor* deletion alone did not result in axon regeneration (*Figure 4—figure supplement 1*); it significantly decreased AKT3-induced axon regeneration (*Figure 6A,B*). The two

best-characterized substrates of mTORC1, S6K1 and 4E-BP, are necessary for axon regeneration (*Yang et al., 2014*; *Hu, 2015*), we tested whether the dominant negative mutant of S6K1 (S6K1-DN) and constitutively active mutant of 4E-BP1 (4E-BP1-4A) also inhibit AKT3-induced axon regeneration. Similar to *Rptor* deletion, S6K1-DN and 4E-BP1-4A also significantly decreased AKT3-induced axon regeneration (*Figure 6A,B*), which further confirms the essential role of mTORC1 in AKT-induced axon regeneration. Since we found that mTORC1 is necessary but that mTORC2 is inhibitory for AKT-induced axon regeneration, we asked what the combined effect would be of blocking both mTORC1 and mTORC2 by deleting *Mtor* itself. We again injected AAV-AKT3 and AAV-Cre together into one eye of *Mtor* floxed mice and compared the ON regeneration to that obtained by injecting AAV-AKT3 alone into the contralateral eye. Deletion of *Mtor* itself did not result in axon regeneration (*Figure 4—figure supplement 1*) but significantly inhibited AKT3-induced axon regeneration (*Figure 6A,B*). This result is additional evidence for mTORC1's essential role in axon regeneration.

Consistent with its key role in protein synthesis, mTORC1 inhibition by deletion of *Rptor* or *Mtor*, or by overexpression of S6K1-DN and 4E-BP1-4A, significantly decreased the AAV-AKT3 expression (*Figure 6C,D*). However, RGC survival was not changed by these manipulations (*Figure 6E,F*), which suggests that the low level of AAV-AKT3 expression in these conditions is sufficient for its neuroprotection function in retina. The inhibitory effect of *Rptor/Mtor* KO or S6K1-DN mutant on pS6 levels (*Figure 6C–F*) implies that downregulation of mTORC1 activity is responsible for the decreased axon regeneration.

## GSK3β phosphorylation and inhibition are necessary for AKT3-induced axon regeneration

We next investigated the influence of another substrate of AKT, GSK3β. GSK3β has been suggested to negatively regulate mammalian peripheral (*Saijilafu et al., 2013*; *Zhang et al., 2014a*) and CNS axon regeneration (*Dill et al., 2008*), although contradictory result has also been reported (*Gobrecht et al., 2014*). To definitively determine the role of GSK3β in CNS axon regeneration, we used AAV-GSK3β-S9A to express a GSK3β mutant that cannot be phosphorylated and inhibited by AKT. GSK3β-S9A significantly blocked AKT3-induced axon regeneration (*Figure 7A,B*). Combining *Rptor* deletion with GSK3β-S9A overexpression blocked both mTORC1 activation and GSK3β inhibition, which almost totally prevented AKT3-induced axon regeneration (*Figure 7A,B*). These results suggest that phosphorylation and inhibition of GSK3β and activation of mTORC1 are two parallel signal pathways downstream of AKT3 that act together to influence axon regeneration.

## *Gsk3b* deletion alone is sufficient for axon regeneration and also acts synergistically with mTORC1 and AKT

To determine whether GSK3β inhibition is sufficient for axon regeneration, we injected AAV-Cre into the eyes of *Gsk3b* floxed mice to delete *Gsk3b* specifically in RGCs. As *Figure 7C,D* shows, *Gsk3b* KO alone induced a small but significant amount of axon regeneration, in dramatic contrast to the deletion of *Gsk3a*, which yielded no axon regeneration. Deletion of both *Gsk3a* and *Gsk3b* did not have an additive effect on axon regeneration (*Figure 4—figure supplement 1*). Since both AKT effectors, mTORC1 and GSK3β, are necessary for axon regeneration and either activation of mTORC1 effector S6K1 (*Yang et al., 2014*) or *Gsk3b* deletion is sufficient to promote axon regeneration, we next tested whether these two pathways have an additive effect. This experiment showed enhancement of axon regeneration when S6K1 constitutively active mutant (S6K1-CA) was expressed in *Gsk3b* KO mice (*Figure 7C,D*), consistent with the idea that mTORC1 and GSK3β act in parallel, synergistic pathways downstream of AKT for axon regeneration.

Since blocking AKT-S473 phosphorylation increased GSK3β-S9 phosphorylation/inactivation (*Figure 5B,G*), we speculated that WT AKT alone only partially inhibits GSK3β and that AKT3 activation together with *Gsk3b* deletion would further increase axon regeneration. Indeed, regenerating axons were more numerous and longer in *Gsk3b* KO mice injected with AAV-AKT3 (*Figure 7E,F*), indicating incomplete inhibition of GSK3β by WT AKT3 and/or AKT-independent GSK3β activity, which acted additively with AKT-dependent GSK3β activity on axon regeneration. Consistent with this inference, AKT-independent GSK3β inactivation has been recognized in peripheral axon regeneration (*Zhang et al., 2014a*).

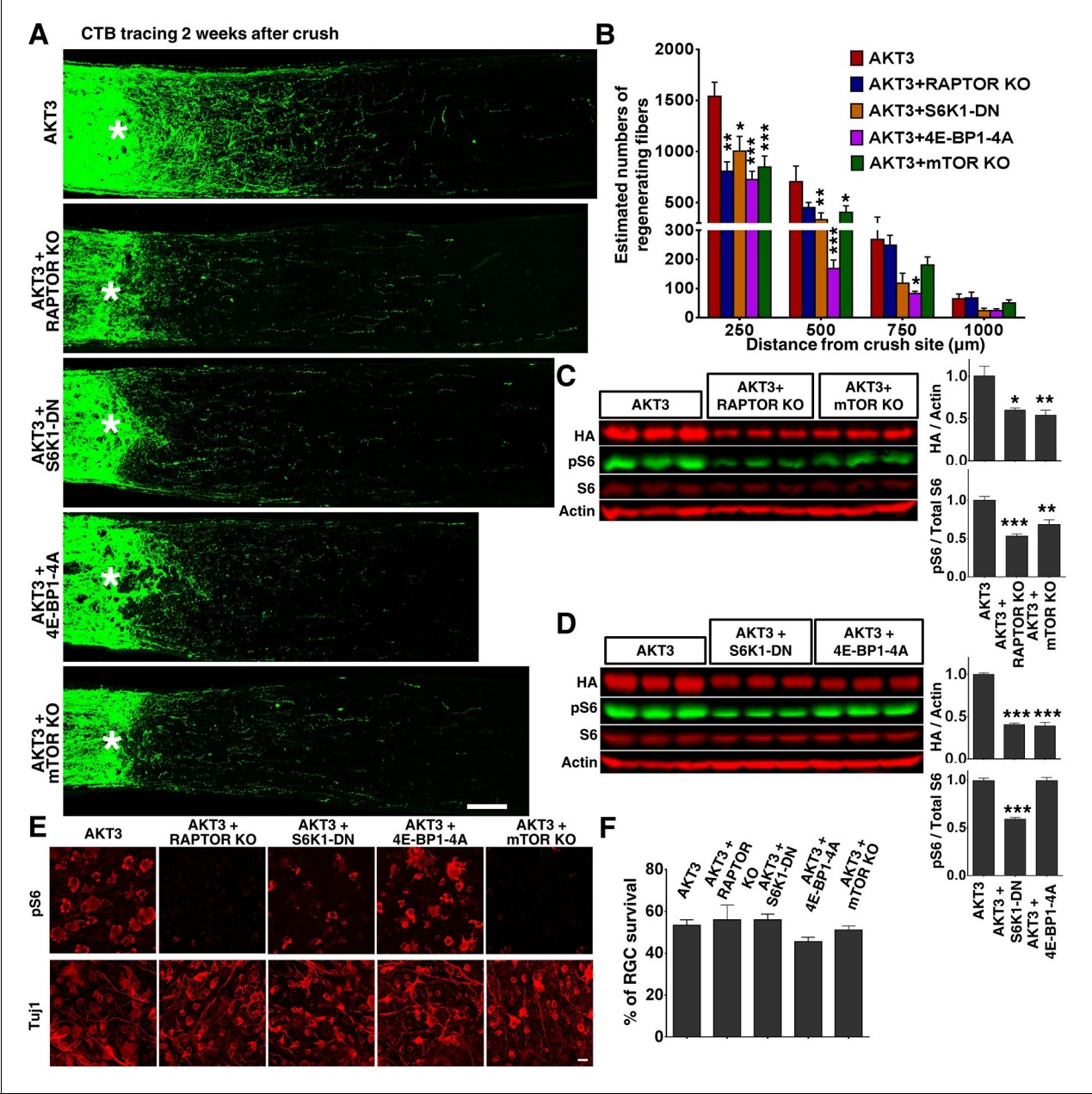

**Figure 6.** mTORC1 and its downstream effectors are essential for AKT3-induced axon regeneration. (**A**) Confocal images of ON longitudinal sections showing regenerating fibers labeled with CTB 2 weeks after ON crush. Scale bar, 100 μm. *crush site. (**B**) Quantification of regenerating fibers at different distances distal to the lesion site. *p<0.05, **p<0.01, ***p<0.001. Data are presented as means ± s.e.m, n=8–10. (**C, D**) Western blots of retina lysates from three biological replicates showing expression levels of HA-AKT3, and phosphorylation levels of S6. (**E**) Confocal images of flat-mounted retinas showing pS6 levels and Tuj1 positive RGCs, 2 weeks after ON crush. Scale bar, 20 μm. (**F**) Quantification of surviving RGCs, represented as percentage of Tuj1 positive RGCs in the injured eye, compared to the intact contralateral eye. Data are presented as means ± s.e.m, n=8–14.

Although AKT3 levels were variable after *Gsk3b* and/or *Rptor* manipulation (*Figure 7—figure supplement 1A,B*), RGC survival induced by AKT3 was unchanged (*Figure 7—figure supplement 1C,D*), indicating sufficient AKT3 expression in retina for neuroprotection. In fact, even though expression of AKT3 was also lower in *Gsk3b* KO mice, axon regeneration was enhanced (*Figure 7E, F*). Thus, the changes in axon regeneration depend on signaling cross-talk and alterations of pS6 and pGSK3β-S9 (*Figure 7—figure supplement 1C*), but not on the absolute levels of AKT3.

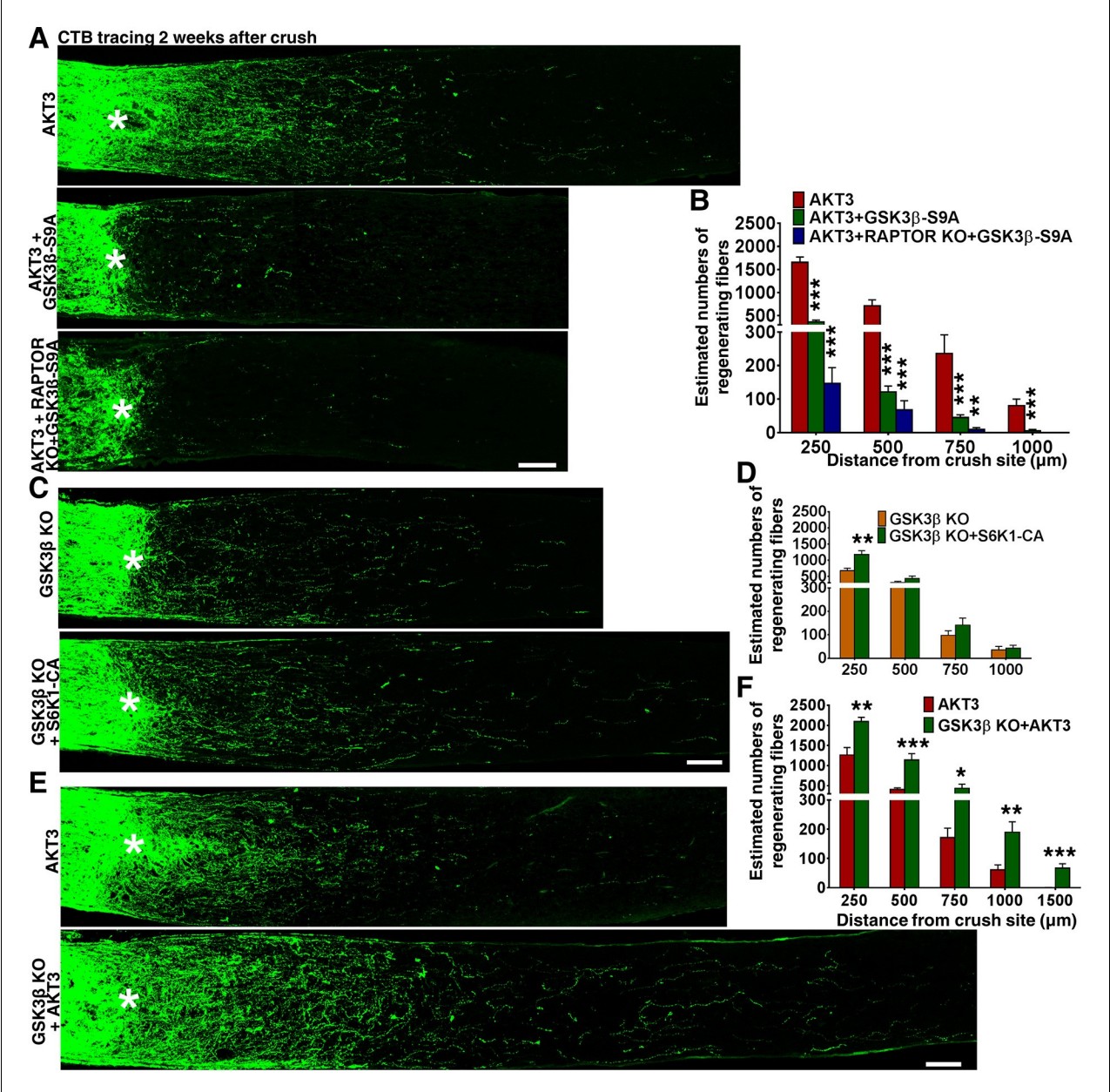

**Figure 7.** GSK3β phosphorylation and inhibition by AKT is necessary and sufficient for axon regeneration. (A,C,E) Confocal images of ON longitudinal sections showing regenerating fibers labeled with CTB 2 weeks after ON crush. Scale bar, 100 μm. *crush site. (B,D,F) Quantification of regenerating fibers at different distances distal to the lesion site. *p<0.05, **p<0.01, ***p<0.001. Data are presented as means ± s.e.m, n=8–15.

The following figure supplement is available for figure 7:

**Figure supplement 1.** AKT3 expression and RGC survival after GSK3β manipulation.

## Discussion

Although AKT has been linked with axon growth in vitro (*Shi et al., 2003*; *Jiang et al., 2005*; *Markus et al., 2002*) and axon regeneration in vivo (*Song et al., 2012b*; *Namikawa et al., 2000*; *Kim et al., 2011a*), there is surprisingly little information about the specificity of AKT isoforms and the role of different phosphorylation sites of AKT (T308 and S473) in axon regeneration. In contrast to AKT1 and AKT2, which are widely expressed, AKT3 is the predominant isoform in brain (*Easton et al., 2005*). Deletion of AKT1 reduces whole body size and deletion of AKT2 results

in diabetes-like syndrome (*Cho et al., 2001a*; *Cho et al., 2001b*). Deletion only of AKT3 specifically reduces brain size (*Easton et al., 2005*), indicating a specific role of AKT3 in regulation of CNS growth. Interestingly, AKT2 and AKT3, but not AKT1, regulate survival and growth of cultured hippocampal neurons (*Diez et al., 2012*). Our analysis of the expression levels of the three AKT isoforms in RGCs and their distinct roles in axon regeneration provides additional in vivo evidence of the unique properties of AKT3 in CNS: AKT3 activation promotes significantly greater RGC survival and ON regeneration than AKT1, presumably through its unique ability to activate mTORC1 (higher pS6) in retina. This is also true in brain as AKT3 KO, but not AKT1 deletion, decreases pS6 significantly (*Easton et al., 2005*). AKT3 may also selectively activate unknown, neuronal-specific signaling molecules. The partial functional redundancy of AKT isoforms, however, makes identification of these molecules difficult (*Dummler and Hemmings, 2007*).

We found that mTORC1 inhibition (*Rptor* or *Mtor* deletion, S6K1-DN or 4E-BP1-4A over-expression) decreased AKT3 expression (*Figure 6C,D*). Decreased AKT3 by itself may not account for reduced axon regeneration, however, because the remaining amounts enabled RGCs to survive (*Figure 6E,F*). Although we cannot exclude the possibility that the decrease in AKT3 contributed to the reduced axon regeneration, we consider the significant decreases in mTORC1 signaling and protein synthesis to be the main reason for the inhibition of axon regeneration. This notion receives additional support from our observation of other conditions in which axon regeneration was enhanced despite downregulation of AKT3 (*Figure 4C,D* and *Figure 7E,F*), which suggests that high levels are not required for axon regeneration. Based on our previous demonstration of the necessary role of 4E-BP and S6K1 (*Yang et al., 2014*),we therefore conclude that mTORC1 is critically important for axon regeneration and that the effects on axon regeneration are due to the signaling alterations, rather than to the changed levels of AKT expression per se.

Our observation of enhanced axon regeneration by AKT3-S472A mutant and *Rictor* KO suggests that pS473 either inhibits phosphorylation of T308 directly or allows AKT to activate distinct substrates that are different from those of pT308, which are inhibitory for axon regeneration. Our finding that pS473 enhances, rather than inhibits, phosphorylation of T308, implies that different substrates or differentially regulated substrates by pT308 and pS473, must contribute to the difference in axon regeneration. The inhibitory effect of mTORC2 and pAKT-S473 on GSK3β phosphorylation is striking and very interesting. The significantly increased pGSK3β-S9 after blocking mTORC2 and S473 phosphorylation suggests that GSK3β is one of the AKT effectors that are differentially regulated by pAKT-T308 and pAKT-S473. Although previous in vitro study showed unchanged GSK3β phosphorylation after *Rictor* KO and reduced S473 phosphorylation (*Guertin et al., 2006*), the liver or muscle-specific *Rictor* deletion increases GSK3β phosphorylation (*Kumar et al., 2008*; *Yuan et al., 2012*). The results of our studies using GSK3β-S9A mutant and *Gsk3b* KO mice provide additional evidence of the inhibitory role of GSK3β in axon regeneration and definitively resolve the contradictory information in the literature (*Christie et al., 2010*; *Abe et al., 2010*; *Saijilafu et al., 2013*; *Dill et al., 2008*; *Gobrecht et al., 2014*). In vitrokinase assay will be necessary to definitively demonstrate the roles of pAKT-T308 and pAKT-S473 in GSK3β-S9 phosphorylation, We have shown that pAKT-T308 and pAKT-S473 regulate the inhibition and phosphorylation of retinal GSK3β in opposite directions. This process acts in parallel with another AKT downstream effector, mTORC1, to regulate CNS axon regeneration (*Figure 8*). Interestingly, downstream effectors that are specific to mTORC2-pAKT-S473 but not to pAKT-T308 have been proposed (*Guertin et al., 2006*; *Yang et al., 2006*; *Jacinto et al., 2004*; *2006*; *2011*), consistent with the notion that pAKT-S473 regulates β-cell proliferation whereas pAKT-T308 controls β-cell size (*Gu et al., 2011*; *Hashimoto et al., 2006*).

The balance between pro-regeneration pT308 and anti-regeneration pS473 forms of AKT appears to be critical. Our observation that *Rictor* deletion induces more potent axon regeneration with a smaller decrease in pS473 and pT308 than AKT3-S472A mutant (*Figure 5*) may indicate a better therapeutic strategy for manipulating AKT signaling to promote axon regeneration. It is also worth noting that, because *Rictor* deletion did not totally block AKT-S473 phosphorylation, other kinases in addition to mTORC2 may contribute in retina. Moreover, because other kinases have been suggested to phosphorylate GSK3β-S9 in addition to AKT (*Tsujio et al., 2000*; *Fang et al., 2000*; *Armstrong et al., 2001*), it will be important to investigate whether these kinases are also regulated by mTORC2 in a similar way as AKT. Since mTORC2 and pAKT-S473 are necessary for PTEN deletion-induced tissue overgrowth in drosophila eyes (*Hietakangas and Cohen, 2007*) and

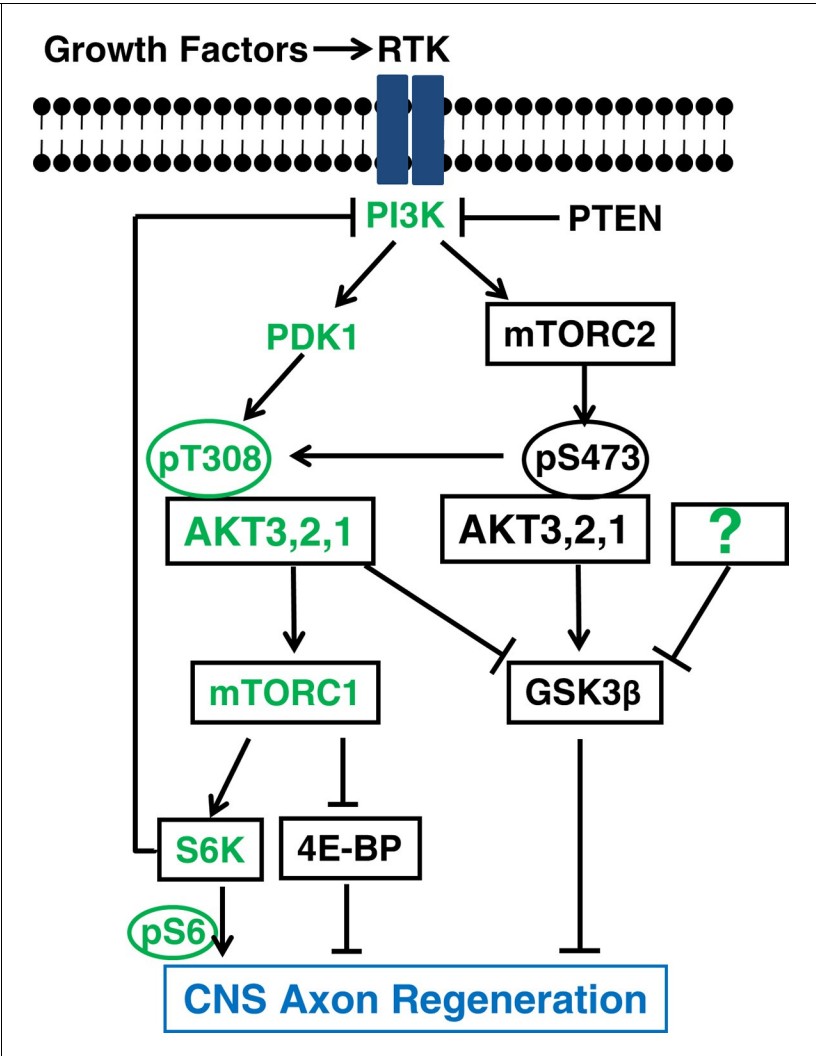

**Figure 8.** A schematic illustration depicting the interplay between AKT, mTORC1/2 and GSK3β in CNS axon regeneration. The predominant isoform of AKT in brain and retina, AKT3, generates the most robust axon regeneration. Its function in axon regeneration is positively regulated by the PI3K/PDK1 pathway through phosphorylation of T308 (T305 in AKT3), and negatively regulated by the PI3K/mTORC2 pathway through phosphorylation of S473 (S472 in AKT3), through at least partially regulation of GSK3β phosphorylation and inhibition. Both AKT downstream effectors, activation of mTORC1 and phosphorylation/inhibition of GSK3β, synergistically promote axon regeneration; inhibition of GSK3β alone is also sufficient for axon regeneration. The question mark represents other upstream regulators of GSK3β that may also promote axon regeneration. Green color-coated molecules are pro-axon regeneration and dark color-coated molecules are anti-axon regeneration.

development of prostate cancer in mouse (*Guertin et al., 2009*), targeting/blocking mTORC2 may allow us to boost PTEN/AKT's regeneration-promoting effect while at the same time minimizing its deleterious tumorigenic effect.

Apparently, neuronal survival is a prerequisite for axon regeneration. But we and others did not find that increased neuron survival was invariably linked to proportionately greater axon regeneration (*Benowitz et al., 2015*). This is consistent with findings in other systems. For example, most corticospinal neurons exhibit long term survival after transection in the spinal cord (*Nielson et al., 2010*; *2011*), but they fail to regenerate axons (*Schwab and Bartholdi, 1996*; *Goldberg et al., 2002b*; *Fitch and Silver, 2008*). The 20% of RGCs that normally survive ON crush in mice can be increased significantly by inhibition of apoptosis, deleting tumor suppressor genes or by manipulating ER stress pathways, but these manipulations do not necessarily induce ON regeneration (*Park et al., 2008*; *Hu et al., 2012*; *Goldberg et al., 2002a*). This observation indicates that axon regeneration requires neuronal intrinsic growth stimulators that are distinct from neuronal

surving factors. Thus we consistently found that, although manipulation of mTOR complexes and GSK3β significantly changed axon regeneration, RGC survival induced by AKT remained the same. We could not exclude the possibility that changing RGC survival contributed to a change in axon regeneration, but no convincing evidence proves a direct causative relationship between these two events. The available evidence, therefore, supports the idea that the intrinsic signaling events after AKT activation and the involvement of its upstream or downstream signaling effectors are directly related to intrinsic growth control of neurons, and that these signaling pathways are distinct from or overlap only partially, signaling necessary for survival. There are far fewer regenerating axons than surviving RGCs, however, suggesting that only a small percentage of RGCs are regenerating and different subtypes of RGCs have different regeneration abilities (*Duan et al., 2015*). In-depth understanding of the mechanisms of this difference will be required to maximize RGC axon regeneration.

Increasing evidence has demonstrated the importance of localized protein synthesis in peripheral axon regeneration (*Willis and Twiss, 2006*; *Jung et al., 2012*; *Perry and Fainzilber, 2014*). Intra-axonal translation has recently been demonstrated in mature mouse hippocampus (*Baleriola et al., 2014*) and, more intriguingly, certain mRNA species and additional components of translation machinery, including pS6 and 4E-BP1, have been detected in regenerating axons in rat spinal cord (*Kalinski et al., 2015*). Since we also observed that regeneration-promoting WT AKTs and AKT-S473A mutant were localized in ON whereas non-regeneration AKT mutants were excluded from ON, it will be very intriguing to investigate the significance of axonal AKT activation in CNS axon regeneration, especially its effect on axonal protein synthesis.

In summary, our genetic manipulations in RGCs have established that the activation of mTORC1 and inhibition of GSK3β are two critical pathways downstream of AKT that act in parallel and synergistically to promote CNS axon regeneration (*Figure 8*). The opposite effects of mTORC1 and mTORC2 on axon regeneration suggest that a balancing mechanism exists downstream of the critical growth-promoting signal PI3K and that AKT integrates both positive and negative signals through phosphorylation of T308 and S473 and their specific roles in downstream effectors mTORC1 and GSK3β to control CNS axon regeneration (*Figure 8*). Interestingly, mTORC1/S6K also functions as feedback inhibition of PI3K signaling (*Laplante and Sabatini, 2012*), which also balances AKT activation and axon regeneration (*Yang et al., 2014*). Thus, it is reasonable to expect that the increased understanding of the complicated cross-regulation and feedback-control mechanisms presented by our studies will eventually lead to safe and effective therapeutic strategies for CNS injury.

## Materials and methods

### Mice

*Rictor*^flox/flox, *Rptor*^flox/flox, *Mtor*^flox/flox and RiboTagmice with C57BL/6 background and C57BL/6 WT mice were purchased from Jackson Laboratories (Bar Harbor, Maine). *Gsk3b*^flox/flox and *Gsk3a*^flox/flox mice with C57BL/6 background were originally developed by Dr. Jim Woodgett (*Doble et al., 2007*; *Patel et al., 2008*) and were acquired from Dr. Thomas Force. We crossed them to generate *Gsk3a/b*^flox/flox mice. All experimental procedures were performed in compliance with animal protocols approved by the IACUC at Temple University School of Medicine. For all surgical and treatment comparisons, control and treatment groups were prepared together in single cohorts, and the experiment repeated at least twice.

### Constructs

pcDNA3-Myr-HA-AKT1 (#9008), pcDNA3-Myr-HA-AKT2 (#9016), pcDNA3-Myr-HA-AKT3 (#9017) and pcDNA3-HA-GSK3β-S9A (#14754) were obtained from Addgene. We used overlap PCR method to produce AKT3 mutants AKT3-T305A, AKT3-S472A, AKT3-T305A/S472A (AKT3-AA), AKT3-K177M (AKT3-KD) and subcloned the WT AKT1-3 and AKT3 mutants into an AAV backbone that contained CBA promoter with Myr-3xHA tag at the N-terminus to get AAV-Myr-3HA-AKT1-3 and AKT3 mutants. We generated AAV-3HA-GSK3β-S9A similarly and AAV-Cre, AAV-S6K1-DN, AAV-S6K1-CA and AAV-4E-BP1-4A were described before (*Yang et al., 2014*).

## AAV production

The detailed procedure has been described previously (*Hu et al., 2012*; *Yang et al., 2014*). Briefly, AAV plasmids containing the transgenes were co-transfected with pAAV2 (pACG2)-RC triple mutant (Y444, 500, 730F) (*Petrs-Silva et al., 2011*; *Wang et al., 2014*; *Zhang et al., 2014b*) and the pHelper plasmid (Stratagene, La Jolla, California) into HEK293T cells. 72 hr after transfection, the cells were lysed to release the viral particles, which were precipitated by 40% polyethylene glycol and purified by cesium chloride density gradient centrifugation. The fractions with refractive index from 1.370 to 1.374 were taken out for dialysis in MWCO 7000 Slide-A –LYZER cassette (Pierce, Thermo Fisher Scientific, Waltham, Massachusetts) overnight at 4°C. The AAV titers that we used for this study were in the range of 1.5–2.5 x $10^{12}$ genome copy (GC)/ml determined by real-time PCR.

## Intravitreal injection and ON crush

Mice were anesthetized by xylazine and ketamine based on their body weight (0.01 mg xylazine/g +0.08 mg ketamine/g). For each AAV intravitreal injection, a micropipette was inserted into the peripheral retina of 3 week-old mice just behind the ora serrata, and advanced into the vitreous chamber so as to avoid damage to the lens. Approximately 2 µl of the vitreous was removed before injection of 2 µl AAV into the vitreous chamber. ON crush was performed 2 weeks following AAV injection: the ON was exposed intraorbitally and crushed with a jeweler's forceps (Dumont #5; Fine Science Tools, Forster City, California) for 5 s approximately 0.5 mm behind the eyeball. Care was taken not to damage the underlying ophthalmic artery. Eye ointment containing neomycin (Akorn, Somerset, New Jersey) was applied to protect the cornea after surgery.

## RGC axon anterograde tracing

2 µl of cholera toxin β subunit (CTB) conjugated with fluorescence Alexa- 488 (2 µg/µl, Invitrogen) was injected into the vitreous chamber 2 days before sacrificing the animals to label the regenerating axons in the optic nerve. Animals were sacrificed by $CO_2$ and fixed by perfusion with 4% paraformaldehyde in cold PBS. Eyes with the nerve segment still attached were dissected out and postfixed in the same fixative for another 2 hr at room temperature. Tissues were cryoprotected through increasing concentrations of sucrose (15%-30%) and optimal cutting temperature compound (OCT) (Tissue Tek, Sakura Finetek, Torrance, California). They were then snap-frozen in dry ice and serial longitudinal cross-sections (8 µm) were cut and stored at –80°C until processed.

## Immunohistochemistry of flat-mount retina

Retinas were dissected out from 4% PFA fixed eyes and washed extensively in PBS before blocking in staining buffer (10% normal goat serum and 2% Triton X-100 in PBS) for 30 min. Mouse or rabbit neuronal class ß-III tubulin (clone Tuj1, 1:500; Covance, Conshohocken, Pennsylvania), rat HA (clone 3F10, 1:200, Roche, Basel, Switzerland), phospho-S6-Ser240/244 antibody (1:200, #5364, Cell Signaling, Danvers, Massachusetts), phospho-AKT-Ser473 (1:200; #4058, Cell Signaling) and phospho-GSK-3β (Ser9) (1:100; #9323, Cell Signaling) were diluted in the same staining buffer. Floating retinas were incubated with primary antibodies overnight at 4°C and washed three times for 30 min each with PBS. Secondary antibodies (Cy2, Cy3 or Cy5-conjugated) were then applied (1:200; Jackson ImmunoResearch, West Grove, Pennsylvania) and incubated for 1 hr at room temperature. Retinas were again washed three times for 30 min each with PBS before a cover slip was attached with Fluoromount-G (Southernbiotech, Birmingham, Alabama).

## Western blot

Retinas were dissected out from ice-cold PBS perfused eyes and homogenized and lysed in RIPA buffer (50 mM Tris HCl pH8.0, 150 mM NaCl, 1% NP-40, 0.5% sodium deoxycholate, 0.1% SDS, 5 mM sodium pyrophosphate, 10 mM sodium fluoride, 1 mM sodium orthovanadate, protease inhibitors cocktail) on ice for 30 min. The homogenates were centrifuged at 12,000 g for 20 min; supernatants were subjected to electrophoresis with 10% SDS-PAGE. After gel-transference, nitrocellulose membranes were blocked with Odyssey blocking buffer (LI-COR, Lincoln, Nebraska) for 1 hr before incubation with primary antibody at 4°C overnight. After washing three times for 10 min with PBS, the membranes were incubated with secondary antibodies (IRDye 680RD goat-anti-mouse IgG or IRDye 800CW goat-anti-rabbit IgG, LI-COR) at room temperature for 1 hr. The membranes were

then washed three times for 10 min with PBS and scanned with Odyssey CLx (LI-COR). The images were analyzed with Image Studio (LI-COR). The primary antibodies used were mouse HA (clone 16B12, 1:2000, Covance), mouse Anti-β-Actin(clone AC-15, 1:2000, Sigma, St Louis, Missouri), and antibodies from Cell Signaling: rabbit phospho-AKT Thr308 (1:500, #2965), rabbit phospho-AKT Ser473 (1:1000, #4058), rabbit phospho-AKT2 Ser474 (1:1000, #8599), rabbit phospho-S6 Ribosomal Protein Ser240/244 (1:1000, #5364), rabbit phospho-GSK-3β Ser9 (1:500; #9323), mouse S6 Ribosomal Protein (1:500, #2317) and mouse GSK-3β (1:500, #9832).

## In situ hybridization

After adult 8-week old mice were perfused with ice-cold 4% PFA/PBS, eyeballs were dissected out and fixed in 4% PFA/PBS at 4°C overnight. The eyeballs were dehydrated with increasing concentrations of sucrose solution (15%-30%) overnight before embedding in OCT on dry ice. Serial cross sections (12 μm) were cut with a Leica cryostat and collected on Superfrost Plus Slides. The sections were washed twice for 10 min in DEPC-treated PBS and permeabilized twice in 0.1% Tween/PBS for 10 min. After blocking at 50°C for 1 hr with hybridization buffer (50% formamide, 5 x SSC, 100 μg/ml Torula Yeast RNA, 100 μg/ml Wheat Germ tRNA, 50 μg/ml heparin and 0.1% Tween in DEPC $H_2O$), the sections were hybridized with 2 μg biotin-labeled antisense probes at 50°C overnight. The sections were washed three times at 55°C for 10 min with hybridization buffer, 0.1% Tween/PBS, and then blocked in PBS blocking buffer containing 0.1% BSA and 0.2% TritonX-100. The hybridized probes were detected by Streptavidin-AP-conjugate (Roche), and revealed by chromogenic substrate NBT/BCIP (Roche). Mouse AKT1, 2, 3 probe sequences were from Allen Brain Atlas (http://mouse.brain-map.org/).

## Counting surviving RGCs and regenerating axons

For RGC counting, whole-mount retinas were immunostained with the Tuj1 antibody, and 6–9 fields were randomly sampled from peripheral regions of each retina. The percentage of RGC survival was calculated as the ratio of surviving RGC numbers in injured eyes compared to contralateral uninjured eyes. For axon counting, the number of CTB- labeled axons was quantified as described previously (*Leon et al., 2000*; *Park et al., 2008*; *Yang et al., 2014*). Briefly, we counted the fibers that crossed perpendicular lines drawn on the ON sections distal to the crush site in increments of 250 μm till 1000 μm, then every 500 μm till no fibers were visible (*Figure 2—figure supplement 1*). The width of the nerve (R) was measured at the point (d) at which the counts were taken and used together with the thickness of the section (t = 8 μm) to calculate the number of axons per $μm^2$ area of the nerve. The formula used to calculate is $\sum a_d = \pi r^2$ * (axon number) / (R*t). The total number of axons per section was then averaged over 3 sections per animal. All CTB signals that were in the range of intensity that was set from lowest intensity to the maximum intensity after background subtraction were counted as individual fibers by Nikon NIS Element R4 software. The investigators who counted the cells or axons were blinded to the treatment of the samples.

## Ribo-IP and RNA-sequencing (RNA-seq)

Three groups of RiboTag mice (8–10 mice/group) were intravitreally injected with AAV2-Cre four weeks before sacrifice and removal of retinas. Ribo-IP was performed according to the published protocol (*Sanz et al., 2009*). Briefly, for each replicate, 10–16 pooled retinas were homogenized and lysed in 1 ml homogenization buffer (50 mM Tris pH7.4, 100 mM KCl, 12 mM MgCl, 1% NP-40, 1 mM DTT, 100 μg/ml cyclohexamide, 1 mg/ml heparin, Protease Inhibitor Cocktail (Sigma) and RNasin Ribonuclease Inhibitor (Promega Corp., Madision, Wisconsin) in RNase-free H2O) on ice for 10 min and centrifuged at 4°C for 10 min at 12,000 g. The supernatant was collected and incubated at 4°C for 4 hr with 10 μg mouse HA antibody, after which 400 μl Dynabeads Protein G (Life Technologies, Frederick, Maryland) were added and incubation continued at 4°C overnight. Dynabeads were washed three times for 10 min with high salt buffer (50 mM Tris pH7.4, 300 mM KCl, 12 mM MgCl, 1% NP-40, 1 mM DTT and 100 μg/ml Cyclohexamide in RNase-free H2O) before RNA extraction with RNeasy Micro Kit (QIAGEN, Hilden, Germany). About 200ng total RNA generated from each group was used for RNA-seq, which was done at the University of Pennsylvania Next-Generation Sequencing Core. Briefly, Ribo-IP RNA samples from three biological replicates went through polyA selection before the generation of strand-specific RNA-seq libraries with Illumina TruSeq Stranded Total Kit and quality assessment with Agilent BioAnalyser and Kapa BioSystems Library

Quant Kit. Pooled libraries that have been individually labeled were sequenced to 100 bp reads from one end of the insert using a HiSeq2000 sequencer. Each library acquired about 25 million reads and 93% of total reads aligned to unique genes in the mouse genome (UCSC mm9) by RUM (*Grant et al., 2011*). The 'raw' data (reads per transcript) were quantile normalized within groups with quantile normalization (GCRMA) to remove non-biological variability.

## Statistical analyses

Data are presented as means ± s.e.m and Student's t-test was used for two-group comparisons and One-way ANOVA with Bonferroni's post hoc test was used for multiple comparisons.

## Acknowledgements

We thank Drs. Michael Selzer, Alan Tessler and Xiaodong Liu for critically reading the manuscript. We are grateful to Dr. Thomas Force for providing *Gsk3a and Gsk3b* floxed mice. Portions of this work were supported by NIH grant EY023295 and EY024932 and Shriners Hospitals for Children research grant #85700 to Y H LM and LY are supported by Postdoctoral Fellowships from Shriners Hospitals for Children.

## Additional information

### Funding

| Funder | Grant reference number | Author |
| --- | --- | --- |
| National Eye Institute | EY023295 | Yang Hu |
| National Eye Institute | EY024932 | Yang Hu |
| Shriners Hospitals for Children | #85700 | Yang Hu |
| Shriners Hospitals for Children | Postdoctoral Fellowship | Linqing Miao Liu Yang |

The funders had no role in study design, data collection and interpretation, or the decision to submit the work for publication.

### Author contributions

LM, LY, YH, Designed the experiments, Performed the experiments and analyzed the data, Prepared the manuscript; HH, FL, Performed the experiments and analyzed the data, Acquisition of data, Contributed unpublished essential data or reagents; CL, Helped with AAV production, Conception and design, Drafting or revising the article, Contributed unpublished essential data or reagents

### Author ORCIDs

Yang Hu, http://orcid.org/0000-0002-7980-1649

### Ethics

Animal experimentation: This study was performed in strict accordance with the recommendations in the Guide for the Care and Use of Laboratory Animals of the National Institutes of Health. All of the animals were handled according to approved institutional animal care and use committee (IACUC) protocols (#4351) of the Temple University School of Medicine.

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
