## [Decision Letter]

Congratulations: we are very pleased to inform you that your article, "mTORC1 Is Necessary But mTORC2 And GSK3β Are Inhibitory For AKT3-Induced CNS Axon Regeneration", has been accepted for publication in *eLife*. The Reviewing Editor for your submission was Nahum Sonenberg and the Senior Editor was K VijayRaghavan. Please note the suggested changes indicated by reviewers #1 and 2 below. If you wish to make these changes, please let us know and we will be happy to consider a final version without returning the manuscript to the reviewers.

Reviewer #1:

In this revised manuscript, the authors have addressed the majority of my concerns. They include quantification of protein expression levels as well as quantification of axon regeneration with different manipulations. The new findings included with the RiboTag mice and the localization of Akt in the axon are interesting. Overall, the manuscript reads better in its revised form, and the observations are presented in a more convincing way.

One control that is important for the new Figure 4—figure supplement 2 is to show the levels of these Akt isoforms/mutants in the cell bodies by western blotting of the corresponding retinas. It is essential to show that the absence of signal in the axon is not due to lower expression levels in the cell body.

Reviewer #2:

The manuscript is overall much improved from its original submission. The changes made in response to the reviewers’ comments have clarified a number of issues. I appreciate the authors including the RGC survival data throughout the work – I think that strengthens the manuscript quite a bit and supports the authors' conclusions for role of AKT3→Torc1 in regeneration. While there certainly are many instances where injured neurons survive but do not regenerate, I do not know of anywhere they regenerate in the absence of survival, Despite the improvement, I think there are a remaining few points of confusion and ambiguity in the data that the authors overlook in interpreting the work. Many of these could be addressed with text changes and some additional controls in the figures.

In Figure 1 there is a remarkable increase in pAKT2-S474 with the myr-AKT2 overexpression. One has to wonder if this is just different affinity between pS474 and pS473 antibodies? However, more concerning is that the authors indicate that transduced myr-AKT3 affords greater regeneration and the endogenous AKT3 is predominant over AKT2. Figure 2 shows that transduced myr-AKT2 provides as good or better regeneration activity than AKT3 and Figure 2—figure supplement 2 shows quite prominent ISH signals and immunoblotting signals for all three isoforms. Granted the RNA-seq data from RGCs shows different loading of AKT1-3 on ribosomes, but I do not think the authors can dismiss the data on regeneration with transduced myr-AKT2 and expression of endogenous AKT2 as they have. This does not detract from the manuscript's findings, but I think the authors should not be misrepresenting their data to the reader (e.g., sentence “The higher pT308 and pS6 levels and more potent axon regeneration induced by AKT3 suggest that its function in retina differs from those of AKT1 and AKT2” is not valid).

The increased regeneration of AKT-S472A vs. WT noted for Figure 4 only appears to be significant for 500 µm distance. Perhaps the authors should note that the remainder show a trend.

In the last paragraph of the subsection “mTORC1 and its downstream effectors are essential for AKT3-induced axon regeneration”, the authors describe the decreased AKT3 expression with S6K-DN and 4E-BP1-4A expression suggesting that the low level of AKT3 is sufficient for (survival) function in the retina. Decrease in AKT3 is also seen for RICTOR KO in Figure 5 but not described. For both of these situations, I think the authors need to be clear that they are referring to levels of the transduced AKT3 not endogenous. Also the authors should indicate that this could reflect a direct effect on protein synthesis as they imply in the aforementioned paragraph, but could also be secondary to altered levels of viral transduction. Both here and in the subsection “GSK3β deletion alone is sufficient for axon regeneration and also acts synergistically with mTORC1 and AKT”, the authors should clarify that they are referring to AKT3 executing a survival function in the cell body rather than distinguishing between soma and axon intrinsic actions of AKT3. Speculation on this difference is more appropriate for the Discussion as the authors have included in this revised version.

Reviewer #3:

The authors have made a comprehensive response to each of the points raised by the reviewers. Most of the points have been fully addressed.

I do not have any further comments.

[Editors’ note: a previous version of this study was rejected after peer review, but the authors submitted for reconsideration. The previous decision letter after peer review is shown below.]

Thank you for submitting your work entitled "mTORC1 Is Necessary But mTORC2 and GSK3β Are Inhibitory For AKT3-Induced CNS Axon Regeneration" for consideration by *eLife*. Your article has been reviewed by three peer reviewers, and the evaluation has been overseen by a Reviewing Editor and K VijayRaghavan as the Senior Editor. Our decision has been reached after consultation between the reviewers. Based on these discussions and the individual reviews below, we regret to inform you that your work will not be considered further for publication in *eLife*.

The authors investigated the signaling pathways playing a role in retinal axon regeneration. It was previously shown that PTEN deletion promotes axon regeneration. Here, the authors study Akt isoforms, mTOR complexes 1 and 2, and downstream S6K and 4EBP using the same in vivo assay. They conclude that Akt3 more than other Akt isoforms is sufficient for regeneration. Similarly, RAPTOR and mTOR are necessary and RICTOR is inhibitory for Akt3-mediated regeneration. Finally, GSK3β and mTORC1 appear to be parallel pathways contributing to retinal axon regeneration.

Overall, the experiments probe an important gap in our understanding of the cellular signaling pathways that mediate retinal axon regeneration. These intrinsic signaling mechanisms likely play a role not only downstream of PTEN deletion but possibly other regeneration-inducing treatments as well. We are turning this paper down because a thorough and complete addressing of the major concerns will take time, and, when done, will result in a substantively different manuscript.

Reviewer #1:

In this manuscript, Miao et al. investigate the signaling pathways playing a role in retinal axon regeneration. It was previously shown that PTEN deletion promotes axon regeneration. Here, the authors study Akt isoforms, mTOR complexes 1 and 2, and downstream S6K and 4EBP using the same in vivo assay. They conclude that Akt3 more than other Akt isoforms is sufficient for regeneration. Similarly, RAPTOR and mTOR are necessary and RICTOR is inhibitory for Akt3-mediated regeneration. Finally, GSK3β and mTORC1 appear to be parallel pathways contributing to retinal axon regeneration.

Overall, the experiments probe an important gap in our understanding of the cellular signaling pathways that mediate retinal axon regeneration. These intrinsic signaling mechanisms likely play a role not only downstream of PTEN deletion but possibly other regeneration-inducing treatments as well.

My main concern about the in vivo experiments is the lack of quantification and controls with respect to the expression of the constructs, especially Akt isoforms and mutants. This raises the question of whether the effects observed are truly due to gene manipulation or due to alterations in the expression of exogenous construction at different levels. Addressing these two important gaps (quantification of cell signaling alterations and quantification of expression levels of tagged proteins) would significantly enhance the impact of this manuscript.

1) If the authors are going to refer to Figure 3 in the Introduction, I suggest that they move it to Figure 1.

2) Figure 1: The authors need to quantify expression levels of HA-tagged AKt isoforms. Also since Ser 235-236 phosphorylation is not specific to S6K, please measure phospho-S6 at Ser 240-244 (antibody available from Cell signaling).

3) Figure 2—figure supplement 1: same criticism as above. Needs quantification of Akt mutants and phospho-S6 at Ser 240-244.

4) Figure 3: It is especially important in this figure to know what the expression levels of Akt3 exogenous construct is in each of these conditions. The effects of mTOR or RAPTOR knockout could simply affect the expression levels of Akt3.

5) Figure 4: similar concerns as above. The authors need to show and quantify the expression levels of Akt3 expression constructs in 4B, E and F.

Reviewer #2:

In this short research report, Miao et al. focus on differential regulation of AKT→mTor pathways for CNS axon regeneration using an optic nerve crush model. The authors convincingly show that membrane targeted AKT proteins increase regeneration, with AKT2 and 3 driving higher levels of growth than AKT1. The differential effects of mTORC1 vs. mTORC2 are intriguing and, if true, have important implications for efforts in the field to augment CNS regeneration using activation of AKT→mTor pathway. Likewise, differential activity of AKT3PT305 vs. AKT3PS472 is a novel finding and a potential advance. Overall, the authors conclude that AKT3PT305→GSK3βPS9 and mTORC1→4EBP1/S6K support CNS axon regeneration while mTORC1→AKT3PS472 blocks or antagonizes axon regeneration. However, I am not convinced that the authors have proven these conclusions.

RGC death following axotomy is well established and the authors indicate that introduction of the AKTmyr isoforms increased RGC survival roughly two fold over non-transduced animals/eyes. The authors sporadically mention survival differences in the text, but show no other data beyond Figure 1. Looking over the post-crush retina images, there is quite variable Tuj1 staining in the supplemental figures. For example, Figure 3—figure supplement 2 shows markedly different numbers of Tuj1+ cells comparing AKT3 only vs. additional AKT3 + raptor KO, S6K-DN, 4EBP1S4A(?) & mTOR KO; Figure 4—figure supplement 2 similarly shows differences in RGC numbers with Tuj1 staining comparing the varying combinations. Considering that the authors only show naïve retina images for the AKT3 mutants and Rictor-KO, I have a very difficult time dissociating regeneration and survival based on the data presented. Obviously, the RGCs need to survive after axotomy to regenerate, but I have concern that some of the decreased regeneration here may relate to differences in survival with the AKT, mTOR, and GSK3β manipulations. The authors need to provide quantitative data for survival across these manipulations and offer clear evidence that effects are regeneration rather than RGC survival to support their conclusions. Going along with this, some of the supplemental data should likely be included in the regular figures for the manuscript.

Similar to the above issue with survival images, the immunoreactivity for pS6K and pGSK3β varies substantially across the same supplemental images and the authors seem to cherry-pick the relevance of these variations.

With the exception of Figure 1—figure supplement 1, the authors do not show any data on wild-type retina/optic nerve. Considering that *eLife* presumably targets a general readership (i.e., rather than selectively neuroscience), the authors need to include data on wild-type animal responses – or better yet, rodents transduced with a control AAV2.

The authors' conclusion that T305 and S472 phosphorylation have distinct outcomes for regeneration is quite intriguing. The schematic in Figure 3 suggests that mTORC2 has a more direct effect on attenuating axon regeneration than antagonizing phosphorylation of AKTT473. If this is the case, then why would the RICTOR KO have no effect on regeneration by itself? Also, when the authors claim that no axon regeneration was detected in mTOR KO, RAPTOR KO, RICTOR KO, against what are the comparisons being performed? It would be useful to see the quantitation of this growth compared to both control viral transduction and wild type AKT3 – likewise, quantitation of the axon growth with the varying AKT3 mutants should be shown.

It is unfortunate that the anti-phospho-AKTT308 does not recognize AKT3PT305, as I think that distinguishing an intramolecular inhibition of T305 phosphorylation vs. indirect mechanisms (including intermolecular between AKT proteins) is an important distinction to make. I do not think it is necessarily needed for the current manuscript, but it would be good to know what efforts the authors made to detect AKT3PT305 as positive data could strengthen the authors' case.

From the work here, as well as previous observations from this group, it is not clear if the effect of modulating the AKT→mTor pathway is mediated in the perikaryon, locally in the axons/growth cones, or both. With increasing evidence for localized protein synthesis in regenerating axons and recent work showing mRNAs and translational machinery (including the mTORC1 targets studied here) in regenerating CNS axons, the authors should acknowledge the possibility of localized effects. The images for HA shown in Figure 1 suggest that AKTmyr isoforms are localizing into at least the proximal axons. Do these also extend to the distal regenerating axons? This also raises the question of whether the regenerating axons shown with CTB labeling are also HA positive (i.e., for those animals expressing AKT isoforms).

Reviewer #3:

The paper looks at AKT and GSKβ effects on regeneration in the optic nerve, assigning roles to various components of the signaling pathway.

1) The authors state at the beginning that they achieved 90% transduction of ganglion cells, but do not appear to have measured this routinely in each case. 90% transduction over the whole retina is difficult to achieve. Moreover, some of the illustrations suggest that the transduction rate is rather more variable and somewhat lower in some of the animals.

2) The increased regeneration is similar to the increase in neuronal survival in Figure 1. Is the increase in axon number a reflection of increase of ganglion cell numbers?

3) The combined effect of active AKT and GSKβknockout in Figure 4 appears to be a significant and robust effect, which is certainly interesting.

4) The effects of the various manipulations 2 and 3 are not large, and would be vulnerable to variable levels of transduction.

5) The reagents used for Figure 3 have overall effects on protein synthesis. To what extent do the authors ascribe their results to specific effects on axon growth rather than through effects on synthesis?

---

## [Author Response]

[Editors’ note: the author responses to the previous round of peer review follow.]

*Reviewer #1: My main concern about the in vivo experiments is the lack of quantification and controls with respect to the expression of the constructs, especially Akt isoforms and mutants. This raises the question of whether the effects observed are truly due to gene manipulation or due to alterations in the expression of exogenous construction at different levels. Addressing these two important gaps (quantification of cell signaling alterations and quantification of expression levels of tagged proteins) would significantly enhance the impact of this manuscript.*We appreciate Reviewer 1’s recognition that our study “probe an important gap[…]”. We thank Reviewer 1 for bring up this insightful request, quantification of expression levels of HA-tagged AKT isoforms/mutants and quantification of cell signaling alterations. We now provide this information, which we believe has substantially improved the manuscript. We realized that immunostaining of flat-mount retina is not sufficient to evaluate subtle differences in expression and alterations in signaling and that this technique will only detect an effect that is very dramatic (all or none). Therefore, we performed Western blots of retina lysates for all the conditions that we studied and quantified the HA-tag levels of AKT isoforms/mutants and signaling (phosphorylation levels of AKT-T308, S473, S6 and GSK3β-S9). We will provide more detail in our responses to individual questions below, but our overall conclusion is that the effects on axon regeneration are indeed due to gene manipulations and corresponding signaling alterations, rather than to different levels of expression of exogenous constructs. Consistent with Reviewer 1’s prediction, these additional Western blot studies significantly enhanced the reliability of our results and therefore the impact of our study. More importantly, the new experiments revealed several new findings. The most significant of these is that: pAKT-T308 and mTORC2-pAKT-S473 differentially regulate GSK3β-S9 phosphorylation, through which axon regeneration is oppositely controlled by pAKT-T308 and mTORC2-pAKT-S473.

1) If the authors are going to refer to Figure 3 in the Introduction, I suggest that they move it to Figure 1.

We agree with the reviewer that it is best not to have a model figure in the middle of the paper and we no longer refer to Figure 3 in the Introduction. Figure 3 now stands by itself as Figure 8 and serves as a summary/model in the last part of the paper, as is customary for most reports.

2) Figure 1: The authors need to quantify expression levels of HA-tagged AKt isoforms. Also since Ser 235-236 phosphorylation is not specific to S6K, please measure phospho-S6 at Ser 240-244 (antibody available from Cell signaling).

We performed Western blots to show comparable expression levels of HA-tagged AKT isoforms in retina. In addition, because anti-pAKT-T308 antibody works well in Western blot, whereas it worked poorly in immunostaining, we were able to find that pAKT-T308 and pS6 are higher in AKT3-retina than AKT1-retina; immunostaining had not definitively revealed this difference before. However, because neither anti-pAKT-T308 nor anti-pAKT-S473 antibodies do not recognize AKT2 well in Western blots, it is difficult to compare AKT2 phosphorylation levels with AKT1 and AKT3. We thank Reviewer 1 for recommending a more reliable antibody for pS6; we repeated all the immunostaining and Western blots of pS6 with phospho-S6 S240/244 antibody, as Reviewer 1 suggested.

3) Figure 2—figure supplement 1: same criticism as above. Needs quantification of Akt mutants and phospho-S6 at Ser 240-244.

We performed Western blots to show comparable expression levels of HA-tagged AKT3 WT and AKT3 mutants in retina and decreased levels of pS6 in the AKT3 mutants (see new Figure 3).

4) Figure 3: It is especially important in this figure to know what the expression levels of Akt3 exogenous construct is in each of these conditions. The effects of mTOR or RAPTOR knockout could simply affect the expression levels of Akt3.

The original Figure 3 is now Figure 6. As Reviewer 1 predicted, mTORC1 inhibition (RAPTOR or mTOR deletion, S6K1-DN or 4E-BP1-4A over-expression) did decrease AKT3 expression as shown by HA levels in Western blot. However, we argue that the decreased AKT3 may not be the reason for decreased axon regeneration, or at least not the sole reason, because the decreased AKT3 did not affect its function in RGC survival, indicating that the level of AKT3 is enough to execute its neuronal function in RGC. Although we cannot totally exclude the possibility that the decreased AKT3 contributes to the decreased axon regeneration, we consider the significant decrease in mTORC1 signaling and protein synthesis to be the main reason for the inhibition of axon regeneration. Consistent with this notion, there are other circumstances in which even though the AKT3 level is downregulated (AKT3 in RICTOR KO or GSK3β KO mice), axon regeneration is enhanced (Figure 5, Figure 7). These findings also support the idea that the level of AKT3 that can induce axon regeneration is not extremely high.

*5) Figure 4: similar concerns as above. The authors need to show and quantify the expression levels of Akt3 expression constructs in 4B, E and F.*The original Figure 4 is now Figure 7 and we include the quantification of AKT3 expression in Figure 7—figure supplement 1. We also observed decreased HA-AKT3 expression after co-expression of GSK3β-S9A mutant or deletion of GSK3β. We don’t know the explanation of this inhibition; it may be due to the mixed-injection of two kinds of AAVs. Again, the decreased AKT3 did not affect RGC survival and showed opposite effects on axon regeneration, depending on activation or inhibition of GSK3β. Thus we conclude that the effects on axon regeneration are due to the signaling alterations, rather than to the level of AKT expression per se.

*Reviewer #2: In this short research report, Miao et al. focus on differential regulation of AKT→mTor pathways for CNS axon regeneration using an optic nerve crush model. The authors convincingly show that membrane targeted AKT proteins increase regeneration, with AKT2 and 3 driving higher levels of growth than AKT1. The differential effects of mTORC1 vs. mTORC2 are intriguing and, if true, have important implications for efforts in the field to augment CNS regeneration using activation of AKT→mTor pathway. Likewise, differential activity of AKT3PT305 vs. AKT3PS472 is a novel finding and a potential advance. Overall, the authors conclude that AKT3PT305→GSK3βPS9 and mTORC1→4EBP1/S6K support CNS axon regeneration while mTORC1→AKT3PS472 blocks or antagonizes axon regeneration. However, I am not convinced that the authors have proven these conclusions.*We appreciate Reviewer 2’s recognition that the findings are “intriguing” and the potential “important implications” of the “novel finding”. We hope that the new experiments and additional data presented in this new submission will convince Reviewer 2 of our conclusions.

*RGC death following axotomy is well established and the authors indicate that introduction of the AKTmyr isoforms increased RGC survival roughly two fold over non-transduced animals/eyes. The authors sporadically mention survival differences in the text, but show no other data beyond Figure 1. Looking over the post-crush retina images, there is quite variable Tuj1 staining in the supplemental figures. For example, Figure 3—figure supplement 2 shows markedly different numbers of Tuj1+ cells comparing AKT3 only vs. additional AKT3 + raptor KO, S6K-DN, 4EBP1S4A(?) & mTOR KO; Figure 4—figure supplement 2 similarly shows differences in RGC numbers with Tuj1 staining comparing the varying combinations. Considering that the authors only show naïve retina images for the AKT3 mutants and Rictor-KO, I have a very difficult time dissociating regeneration and survival based on the data presented. Obviously, the RGCs need to survive after axotomy to regenerate, but I have concern that some of the decreased regeneration here may relate to differences in survival with the AKT, mTOR, and GSK3β manipulations. The authors need to provide quantitative data for survival across these manipulations and offer clear evidence that effects are regeneration rather than RGC survival to support their conclusions. Going along with this, some of the supplemental data should likely be included in the regular figures for the manuscript.*We apologize for the inadequate RGC survival data in the original manuscript. We had collected all the RGC survival data in every condition that we tested for axon regeneration, but we did not present it in the previous version, because we wanted to save space and because we do not consider RGC survival to be a decisive factor for axon regeneration, based on our previous experience and the available literature. We now present all the quantitative RGC survival data in Figure 2, Figure 4—figure supplement 3, and Figure 7—figure supplement 1.

We regret the confusing Tuj1 staining in the original figures. Our monitors did not indicate “markedly different numbers of Tuj1+ cells”, although fluorescence intensity can vary from monitor to monitor, perhaps because of differences in tone calibration. We have put considerable effort into presenting the figures in a consistent fashion and counting the surviving RGCs with double-blinded methods.

We agree with Reviewer 2 that a RGC must survive to be able to regenerate its axon. However, the level of RGC survival does not seem to correlate with the degree of axon regeneration in many of the conditions that we studied. We now include a paragraph in the Discussion which reads:

“Apparently, neuronal survival is a prerequisite for axon regeneration. But we and others did not find that increased neuron survival was invariably linked with proportionately greater axon regeneration (Benowitz et al. 2015). […] There are far fewer regenerating axons than surviving RGCs, however, suggesting that only a small percentage of RGCs are regenerating and different subtypes of RGCs have different regeneration abilities (Duan et al. 2015). In-depth understanding of the mechanisms of this difference will be required to maximize RGC axon regeneration.”

Similar to the above issue with survival images, the immunoreactivity for pS6K and pGSK3β varies substantially across the same supplemental images and the authors seem to cherry-pick the relevance of these variations.

We agree with Reviewer 2’s opinion about the inadequacies of immunostaining experiments and we have included Western blot quantification data, which are more convincing and relevant. We have learned that immunostaining can reliably detect only large differences (e.g., totally blocked, deleted or overexpressed genes), whereas Western blot can show relative quantitatively even relatively subtle signaling differences.

*With the exception of Figure 1—figure supplement 1, the authors do not show any data on wild-type retina/optic nerve. Considering that eLife presumably targets a general readership (i.e., rather than selectively neuroscience), the authors need to include data on wild-type animal responses – or better yet, rodents transduced with a control AAV2.*

We now present WT retina and optic nerve images in Figure 2, Figure 4 and Figure 4—figure supplement 1.

The authors' conclusion that T305 and S472 phosphorylation have distinct outcomes for regeneration is quite intriguing. The schematic in Figure 3 suggests that mTORC2 has a more direct effect on attenuating axon regeneration than antagonizing phosphorylation of AKTT473. If this is the case, then why would the RICTOR KO have no effect on regeneration by itself?

We agree with Reviewer 2’s interpretation, that mTORC2 might not regulate axon regeneration directly since RICTOR KO does not induce axon regeneration. Originally we considered that RICTOR deletion may just not be enough to induce axon regeneration; because we have not proven this notion, we have removed from the new Figure 8 the dotted line that linked mTORC2 directly to axon regeneration in the original Figure 3.

*Also, when the authors claim that no axon regeneration was detected in mTOR KO, RAPTOR KO, RICTOR KO, against what are the comparisons being performed? It would be useful to see the quantitation of this growth compared to both control viral transduction and wild type AKT3 – likewise, quantitation of the axon growth with the varying AKT3 mutants should be shown.*We regret not making it clear that we consider that there is no axon regeneration when the phenotype is similar to WT mice. We now provide data, including WT optic nerve after crush injury in Figure 4 and Figure 4—figure supplement 1, to show the no axon regeneration phenotype of AKT3 mutants and mTOR/RAPTOR/RICTOR deletion and the quantification data.

It is unfortunate that the anti-phospho-AKTT308 does not recognize AKT3PT305, as I think that distinguishing an intramolecular inhibition of T305 phosphorylation vs. indirect mechanisms (including intermolecular between AKT proteins) is an important distinction to make. I do not think it is necessarily needed for the current manuscript, but it would be good to know what efforts the authors made to detect AKT3PT305 as positive data could strengthen the authors' case.

We thank Reviewer 2 for the suggestion to test anti-phospho-AKT-T308 antibody, which indeed works in Western blot. As Reviewer 2 predicted, we confirmed the activation of AKT by overexpression of constitutively active WT AKTs and the inhibition of AKT3 by the AKT3 mutants in Figure 1, Figure 3 and 5. More importantly, we determined that pS473 inhibition by AKT3-S472A mutant or RICTOR KO decreases pT308. These results provide additional confirmation of our hypothesis that the increased axon regeneration induced by blocking S473 phosphorylation is not due to the increase of pT308, but to different downstream effectors or differentially regulated downstream effectors of AKT. Surprisingly, but excitingly, we found that phosphorylation of GSK3β is oppositely regulated by pT308 and pS473 (Figure 5). This finding is extremely important, but was missed from our original manuscript. We therefore greatly appreciate that both Reviewer 1 and Reviewer 2 insisted that we should explore signaling events with Western blots.

*From the work here, as well as previous observations from this group, it is not clear if the effect of modulating the AKT→mTor pathway is mediated in the perikaryon, locally in the axons/growth cones, or both. With increasing evidence for localized protein synthesis in regenerating axons and recent work showing mRNAs and translational machinery (including the mTORC1 targets studied here) in regenerating CNS axons, the authors should acknowledge the possibility of localized effects. The images for HA shown in Figure 1 suggest that AKTmyr isoforms are localizing into at least the proximal axons. Do these also extend to the distal regenerating axons? This also raises the question of whether the regenerating axons shown with CTB labeling are also HA positive (i.e., for those animals expressing AKT isoforms).*This is another case in which Reviewer 2’s suggestion deserves our thanks. As Reviewer 2 suggested, we investigated the translocation of AKT in optic nerve and discovered another surprising distinct phenotype (Figure 4—figure supplement 2): pro-axon regeneration WT AKT isoforms and AKT3-S472A are distributed in optic nerve whereas no-axon regeneration AKT3-KD, AA and T305A mutants are significantly blocked from optic nerve. The mechanism for this difference and the relationship to axon regeneration are presently unclear. However, this important finding, which otherwise would be missed without Reviewer 2’s thoughtful input, together with the confirmation of localization of WT AKT in regenerating axons, open a new avenue to explore in CNS axon regeneration.

Reviewer #3:

The paper looks at AKT and GSKβ effects on regeneration in the optic nerve, assigning roles to various components of the signaling pathway.

*1) The authors state at the beginning that they achieved 90% transduction of ganglion cells, but do not appear to have measured this routinely in each case. 90% transduction over the whole retina is difficult to achieve. Moreover, some of the illustrations suggest that the transduction rate is rather more variable and somewhat lower in some of the animals.*

We have used a mutant AAV capsid plasmid to produce AAV2 that can achieve very high transducing efficiency for RGCs. We previously explained this strategy in the Methods section, and we now also note it in Results. We also now include quantification of percentages of Tuj1+ RGCs expressing HA-tagged AKTs (> 80%). The original claim of 90% is indeed achievable but not always. We agree with Reviewer 3 that there will be variation in in vivo studies. We used the same batch of virus and the same titers across the lines to minimize the variation sufficiently for us to be confident with our results.

2) The increased regeneration is similar to the increase in neuronal survival in Figure 1. Is the increase in axon number a reflection of increase of ganglion cell numbers?

The regenerating axons are far fewer than the surviving RGCs, and we do not think that the variation in RGC survival is the main reason for the change in axon regeneration, although we cannot totally exclude this possibility. We have explained this in detail in our response to Reviewer 2’s similar comment and discuss it in the manuscript.

3) The combined effect of active AKT and GSKβ knockout in Figure 4 appears to be a significant and robust effect, which is certainly interesting.

We appreciate Reviewer 3’s recognition of the significance of this finding. In the revised manuscript we further provide an explanation and exciting evidence, the less complete phosphorylation/inhibition of GSK3β by WT AKT.

4) The effects of the various manipulations 2 and 3 are not large, and would be vulnerable to variable levels of transduction.

We are not sure what “manipulations 2 and 3” refer to. The expression levels for all the manipulations are now documented by Western blot analysis and hopefully our responses to other questions answered this question as well.

5) The reagents used for Figure 3 have overall effects on protein synthesis. To what extent do the authors ascribe their results to specific effects on axon growth rather than through effects on synthesis?

This is a good question; we addressed a similar question in our response to Reviewer 1’s comments.